# Landscape of somatic single nucleotide variants and indels in colorectal cancer and impact on survival

Syed H. Zaidi et al.[#]

Colorectal cancer (CRC) is a biologically heterogeneous disease. To characterize its mutational profile, we conduct targeted sequencing of 205 genes for 2,105 CRC cases with survival data. Our data shows several findings in addition to enhancing the existing knowledge of CRC. We identify *PRKCI*, *SPZ1*, *MUTYH*, *MAP2K4*, *FETUB*, and *TGFBR2* as additional genes significantly mutated in CRC. We find that among hypermutated tumors, an increased mutation burden is associated with improved CRC-specific survival (HR = 0.42, 95% CI: 0.21–0.82). Mutations in *TP53* are associated with poorer CRC-specific survival, which is most pronounced in cases carrying *TP53* mutations with predicted 0% transcriptional activity (HR = 1.53, 95% CI: 1.21–1.94). Furthermore, we observe differences in mutational frequency of several genes and pathways by tumor location, stage, and sex. Overall, this large study provides deep insights into somatic mutations in CRC, and their potential relationships with survival and tumor features.

[#]A list of authors and their affiliations appears at the end of the paper.

Colorectal cancer (CRC) is the third most common cancer and the second leading cause of cancer deaths worldwide[1]. CRC is a complex disease caused by multiple environmental, lifestyle, and genetic risk factors. Exposures to exogenous and endogenous factors, aberrant DNA editing, and defective DNA maintenance cause mutations and epigenetic alterations, which confer cellular transformation and growth, leading to the development of CRC.

Next-generation sequencing (NGS) has identified a diversity of driver mutations in genes and altered signaling pathways in CRC[2–5]. Limitations of these studies are the scarcity of clinical data, as well as the inability to achieve statistical significance due to small sample size. Advances in DNA extraction from archived formalin-fixed paraffin-embedded (FFPE) tissues and sequencing have enabled us to utilize a sizeable collection of CRC cases with available clinical data from the Genetics and Epidemiology of Colorectal Cancer Consortium (GECCO) and the Colon Cancer Family Registry (CCFR). Here, we present a systematic mutation analysis of 2105 CRC cases using targeted, deep sequencing data. We constructed a custom AmpliSeq panel of 205 genes, prioritized from the analyses of CRC mutation datasets and literature review[2,6,7], and conducted targeted deep sequencing on DNA from FFPE tumors and matching normal tissues from five-well-characterized studies. The profiling of mutations in the largest population-based CRC sequencing study to date provides a deep insight into the mutational landscape of CRC and associations with survival.

## Results

**Targeted sequencing**. To construct an AmpliSeq panel, we prioritized a list of 205 genes selected based on analysis of whole exome sequencing data for CRC from TCGA[2,6], and two prospective cohort studies, the Health Professionals' Follow-Up Study (HPFS) and the Nurses' Health Study (NHS)[7], as well as a literature search. We included homopolymer repeats to evaluate microsatellite stability (see Supplementary Methods for panel design). The DNA extraction, sequencing, and mutation calling are described in detail in "Methods" and Supplement sections. We successfully sequenced tumor DNA (from FFPE tissue) and normal DNA (primarily from blood) from 2105 CRC cases recruited across five observational studies that collected survival data, CORSA, OFCCR, SCCFR, CPS-II, and DACHS. The mean sequencing coverage of tumor and normal DNA was 857× and 302×, respectively.

**Frequency and type of mutations and hypermutated tumors**. Among the 2105 CRC tumors, we identified a total of 25,586 synonymous, and 29,947 non-silent somatic mutations (Supplementary Data 1). The non-silent mutations consisted of 19,838 missense mutations (66%), 3152 nonsense mutations (11%), 15 stop losses (0.05%), 541 splice site mutations (2%), 6203 frameshift indels (21%), and 194 in-frame indels (0.7%, Supplementary Fig. 1). Tumor mutation frequency varied substantially across samples (Fig. 1). A total of 19% of all CRC cases were defined as hypermutated (HM) based on the frequency distribution of somatic point mutations from all samples (see "Methods"). As expected, a large fraction of the HM tumors was MSI (67.8%). MSS-HM tumors were enriched for nonsynonymous point mutations in the proofreading exonuclease domain of POLE (Fig. 1). The HM tumors with POLE exonuclease domain mutations exhibited an ultra-hypermutated phenotype whereas this was not seen in HM tumors with POLD1 exonuclease domain mutations (mean number of somatic mutations = 235.6 (sd = 158.2) vs. 144.1 (sd = 80.0), respectively). In HM tumors, recurrent mutations in the exonuclease domains include

P286R/S (3%), V411L (1%), S459F (1%), F367C/S/V (0.8%), P436R/S (0.5%), and A456P (0.5%) in POLE, and R454C/H (1%), E318K (0.8%), R352C (0.5%), R470C/H (0.5%), and V477M (0.5%) in POLD1. In MSS-HM tumors without mutations in the POLE and POLD1 genes (n = 64), we examined other genes on our panel that could affect DNA replication or repair. A subset of these tumors (n = 30), contained mutations in CDK12, ATM, RECQL5, FAN1, NCAPD3, ERCC3, XPC, NTHL1, and passenger mutations in MMR genes (Supplementary Fig. 2). The remaining MSS-HM tumors (n = 34) may carry mutations in other DNA repair genes not included in the current panel. A detailed analysis of the newly identified subset of MSS-HM tumors without non-silent mutations in POLE and POLD1 is described at the end of the "Results" section.

Overall, we observed that HM tumors were less likely to be diagnosed at stage IV than non-hypermutated (NHM) tumors (4% vs. 10%, respectively), and more likely to arise in right-sided CRC (76% vs. 24%). We also observed that CRC-specific survival was significantly more favorable among individuals with HM tumors than among those with NHM CRC (HR = 0.36, 95% CI: 0.24–0.54). This association was consistent regardless of stage at diagnosis or tumor site, and was not impacted by adjustment for these variables. Associations with survival were also consistent among both those with (HR = 0.24, 95% CI: 0.10–0.58) and without somatic POLE exonuclease domain mutations (HR = 0.41, 95% CI: 0.26–0.65).

**Frequently mutated genes**. We defined gene mutations based on the presence of non-silent mutations. As expected, we observed substantial differences in the mutational frequency of genes between NHM tumors and HM tumors (Fig. 2). In NHM, the most frequently mutated genes based on non-silent mutations included APC, TP53, KRAS, SYNE1, PIK3CA, FBXW7, SOX9, RYR1, and SMAD4 (Fig. 2a). These genes also harbored non-silent mutations in the HM tumors, but were mutated at different frequencies in the HM set. Among the HM tumors, SYNE1, RYR1, RNF43, and KMT2D were the most commonly mutated genes. Several of the frequently mutated and some of the less frequent mutated genes in NHM and HM tumors were classified as significantly mutated by MutSigCV (q < 0.1; Fig. 2a). In addition to previous known genes, we identified PRKCI, SPZ1, MUTYH, MAP2K4, FETUB, and TGFBR2 as significantly mutated by MutSigCV, which had not been reported in previous studies[2,4,6,7], suggesting putative driver status of these genes. Validation of 84 mutations from these genes by Sanger sequencing showed 98.8% correct calls.

In analyses across all CRC cases, we examined associations between gene-level mutations and CRC-specific survival, accounting for multiple testing and restricting our analyses to genes with at least 10 CRC deaths in those with non-silent mutations. Median survival time ranged from 60.6 to 194.1 months across the included studies (Supplementary Table 1), and the study-specific proportion of participants who died from CRC ranged from 14.8 to 27.0%. No gene-level mutations were significantly associated with CRC-specific survival after adjusting for age, sex, mutational burden and study (Supplementary Data 2). Similarly, we did not find any gene significantly associated with CRC-specific survival when we stratified the analyses by hypermutation status, although such stratified analyses were based on smaller numbers.

**Alterations in main signaling pathways implicated in CRC**. We conducted a detailed analysis of primary pathways implicated in CRC[2]. A total of 82% of tumors carried non-silent mutations in genes belonging to more than one signaling pathway, though

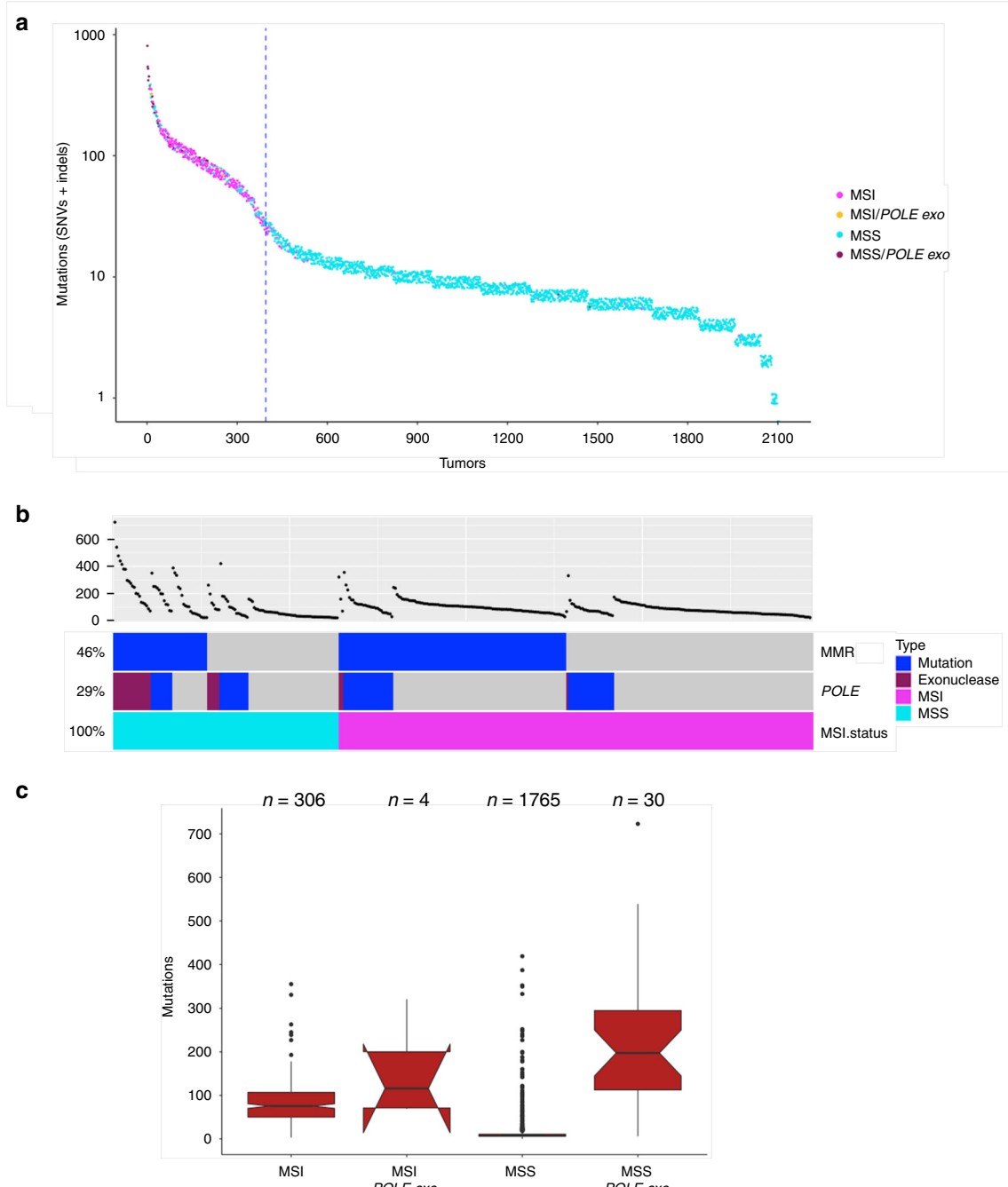

**Fig. 1 Mutation profiling of 2105 colorectal tumors. a** Tumors are sorted based on the number of mutations with each dot indicating mutations in that tumor. Jitter was added to easier visualize overlapping data points. The vertical dotted line separates hypermutated and non-hypermutated tumors (see "Methods"). Tumors with MSI and *POLE* exonuclease domain mutations are frequent in hypermutated tumors. **b** Analysis of hypermutated tumors. Tumors with mutations in DNA mismatch repair genes (MMR: *MLH1*, *MLH3*, *MSH2*, *MSH6*, or *PMS2*), tumors with non-silent non-truncating mutations in *POLE*, and their MSI status are shown. **c** MSI or MSS tumors were examined for the impact of *POLE* exonuclease non-silent non-truncating mutations on overall mutation burden. The boxplots show tumors with and without *POLE* exonuclease (exo) domain mutations and the MSI status. The center line, bounds, and whiskers of the boxplots are median, first and third quartiles, and outliers, respectively. The medians for boxes without overlapping notches are significantly different at the 0.95 confidence level. MSS tumors with mutations in the *POLE* exonuclease domain have significantly higher mutation burden compared to the MSS and MSI tumors without the *POLE* exonuclease domain mutations.

differences in mutation frequencies were observed between HM and NHM tumors (Fig. 2b). In NHM tumors, 77% of tumors have mutations in the WNT/beta-catenin pathway, followed by TP53/ ATM (62%), receptor tyrosine kinases/RAS (RTK/RAS, 50%), transforming growth factor-beta (TGF-beta, 21%), and IGF2/ phosphatidylinositide 3-kinases (PI-3-kinase, 17%) pathways. In

HM tumors, 97% of tumors were mutated in WNT/beta-catenin signaling genes, followed by TGF-beta (80%), RTK/RAS (74%), TP53/ATM (48%), and IGF2/PI-3-kinase (46%) pathways. Contributions of mutated genes in individual pathways are shown in Fig. 2c and Supplementary Fig. 3. Overall, 96% of NHM and 100% of HM tumors displayed at least one non-silent mutation in

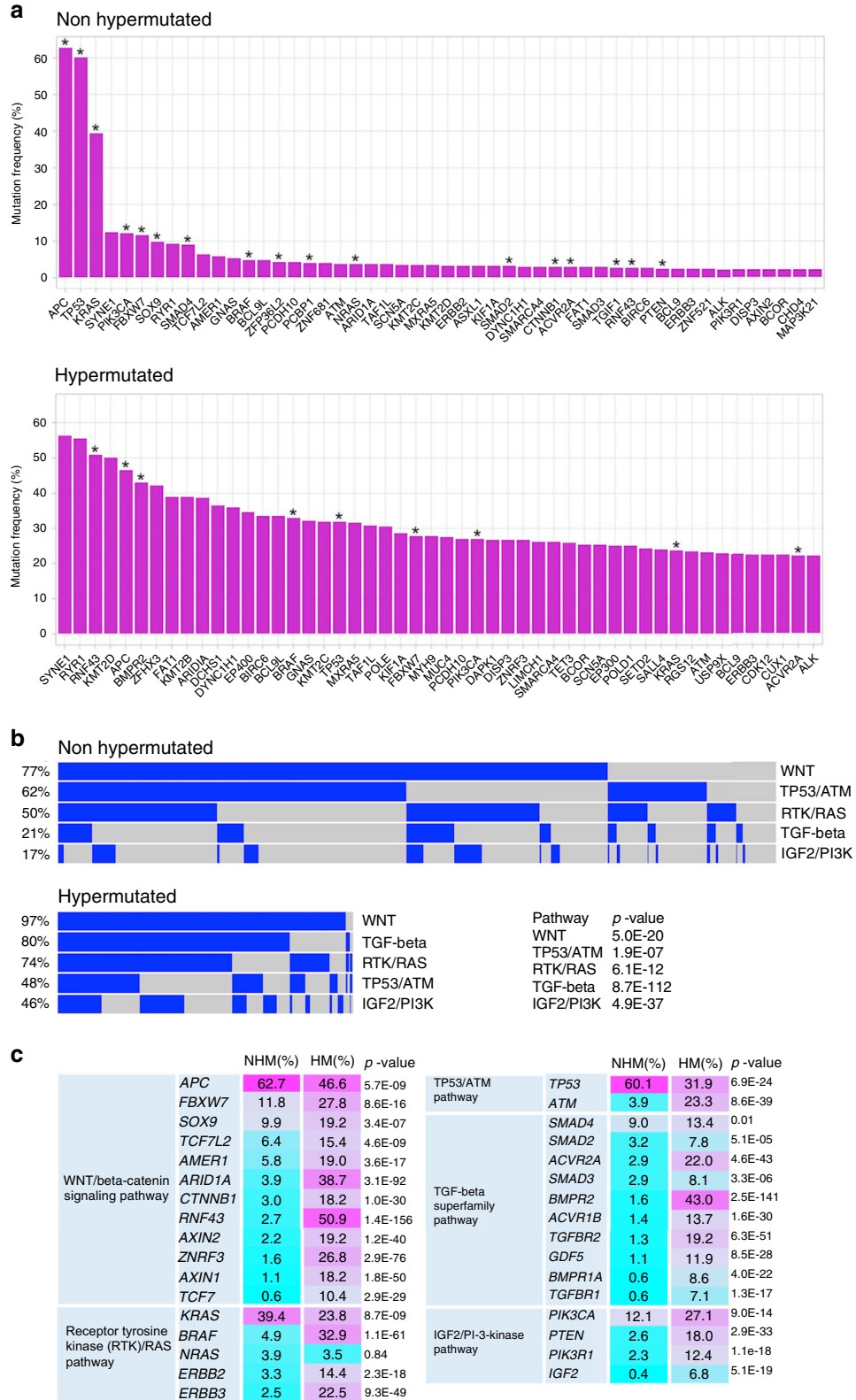

**Fig. 2 Non-silent mutations in commonly mutated genes and altered pathways in colorectal tumors. a** The top 50 mutated genes in non-hypermutated and hypermutated tumors are shown. Significantly mutated genes identified by MutSigCV (*q* < 0.1) are indicated with asterisks. **b** Oncoprint display of alterations in main signaling pathways in non-hypermutated and hypermutated tumors. Contributions of individual genes to pathways are shown in Supplementary Fig. 2. Chi-square test *p*-values show significant differences in mutated pathways between non-hypermutated and hypermutated tumors. **c** Frequencies of mutated genes in pathways in non-hypermutated (NHM) and hypermutated (HM) tumors are shown. Chi-square test *p*-values compare mutation frequencies of genes with non-silent mutations between non-hypermutated and hypermutated tumors.

a gene belonging to the main signaling pathways implicated in CRC. Compared to the NHM, HM tumors had more alterations in multiple pathways.

*WNT/CTNNB1(beta-catenin) signaling pathway*: In the WNT-signaling pathway, *APC* most frequently carried non-silent mutations. Among *APC* mutated tumors, 99% of NHM and 85% of HM tumors harbored truncating mutations occurring within the first 1600 codons, for which truncating mutations are predicted to have the most functional consequences[8] (Supplementary Fig. 4). Approximately 19% of all NHM and 22% of all HM tumors carried two or more non-silent mutations in *APC*.

Among *CTNNB1*-mutated tumors, missense point mutations and in-frame indels in the D32-S45 hotspot region of *CTNNB1* were significantly more frequent among tumors without truncations within the first 1600 codons of APC when compared to tumors with *APC* inactivating mutations in the first 1600 codons ($p$ value $< 1 \times 10^{-5}$, Fisher exact test; (Fig. 3)) suggesting mutual exclusivity. In the tumors with *CTNNB1* hotspot mutations, we observed mutations that alter codons S45 ($n = 17$), T41 ($n = 16$), S37 ($n = 3$), G34 ($n = 4$), S33 ($n = 1$), and D32 ($n = 1$).

We did not observe a significant association between having any WNT-signaling pathway gene mutation and CRC survival (Supplementary Data 2) or between *CTNNB1* hotspot mutations and CRC survival.

*TP53/ATM pathway*: *TP53* was the second most commonly mutated gene in NHM tumors (60%), but less often mutated in HM tumors (32%). In contrast, the *ATM* gene was more frequently mutated in HM tumors (23%) than NHM tumors (4%; Fig. 2 and Supplementary Fig. 3). In *TP53* 76% of the mutations were missense which were predominantly in the DNA binding domain, and 23% of the mutations were truncating which were distributed along the entire length of the protein (Supplementary Fig. 4). All in-frame indels in *TP53* (1%) were found in the DNA binding domain. Approximately 5% of *TP53* mutated tumors carried two or more non-silent mutations in *TP53*.

The presence of a non-silent somatic mutation in *TP53* was associated with modestly poorer CRC survival in all cases combined (HR = 1.27, 95% CI: 1.01–1.59, Supplementary Data 2). When categorizing somatic mutations by deleteriousness based on residual transcriptional activity, the association with CRC-specific survival was more pronounced (Table 1). TP53 residual activity was determined using the International Agency for Research on Cancer (IARC) *TP53* database[9] (See "Methods"). Among individuals with a somatic mutation in *TP53*, those with 0% predicted residual TP53 activity had significantly poorer CRC-specific survival compared to those with mutations causing >5% predicted residual activity or those with no mutations (HR = 1.53, 95% CI: 1.21–1.94) after adjusting for age, sex, mutational burden, and study. This association was primarily driven by results among NHM cases (HR = 1.52, 95% CI: 1.19–1.94), among whom the observed association was modestly impacted by adjustment for stage at diagnosis (HR = 1.36, 95% CI: 1.06–1.95). *TP53* mutations that resulted in minimal residual activity (>0% but <5%) were also associated with a non-statistically significant poorer CRC-specific survival (HR = 1.38, 95% CI: 0.89–2.13). *TP53* mutations were less common in right-sided vs. left-sided CRC, and the proportion of tumors in right-sided CRC decreased with decreasing TP53 residual activity. However, adjustment for tumor site did not impact the observed association of *TP53* mutation status with CRC survival, nor did these associations vary substantially by tumor site.

*Receptor tyrosine kinase (RTK)/RAS pathway*: Among genes analyzed in this pathway, *KRAS* was the most frequently mutated, followed by *BRAF* and *NRAS* in NHM tumors and *BRAF*, *KRAS*, and *ERBB3* in HM tumors (Fig. 2 and Supplementary Fig. 3). As expected, the frequently mutated codons in *KRAS* and *NRAS* were G12, G13, and Q61 and in *BRAF* was V600 (Supplementary Figs. 4 and 5). In the RTK/RAS pathway, we demonstrate that most mutations in *KRAS*, *NRAS*, *BRAF*, *ERBB2*, and *ERBB3* are mutually exclusive (Supplementary Fig. 3).

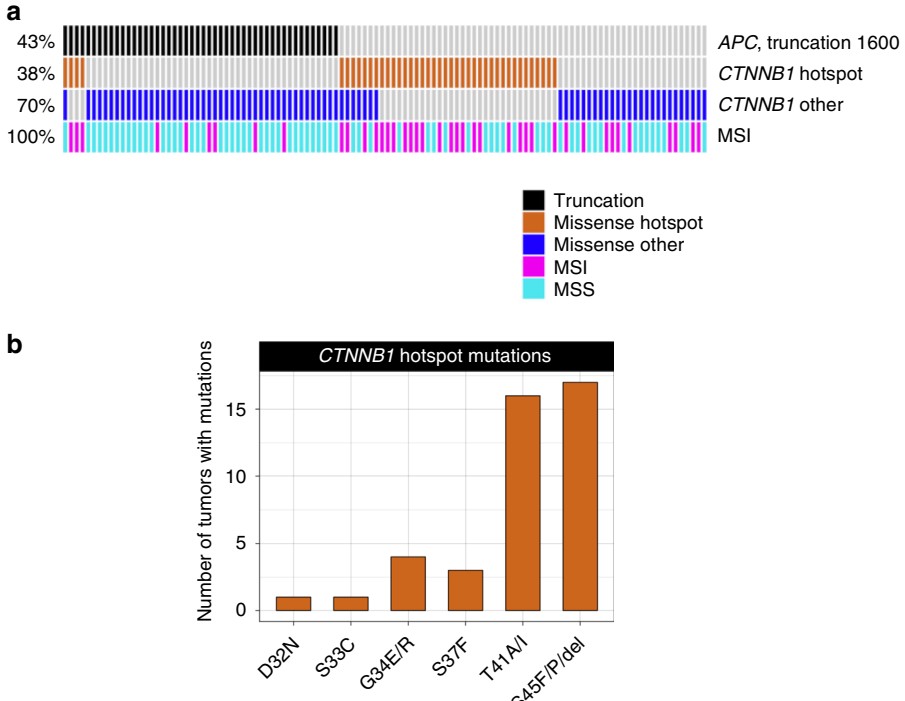

**Fig. 3 Profiling of colorectal tumors with *CTNNB1* missense mutations and in-frame indels. a** *CTNNB1* hot spot mutations affecting codons D32 to S45 are enriched in tumors without APC truncations within the first 1600 codons. The *CTNNB1* hotspot mutations are frequent in tumors with MSI. **b** The number of tumors mutated in the mutation hotspot region of *CTNNB1*. S33, S37, T41, and S45 are the sites of phosphorylation by kinase.

**Table 1 Survival analyses for *TP53* somatic mutations defined by transcriptional activity and stratified by hypermutation status.**

| Mutation group | Hypermutation status | Cases (*n*) | CRC-specific deaths (*n*) | HR | 95% CI[a] | *P* value[a] |
|---|---|---|---|---|---|---|
| >5% residual activity or no mutation | Combined | 1083 | 169 | 1.00 | (Ref) | – |
| | NHM | 806 | 150 | 1.00 | (Ref) | – |
| | HM | 277 | 19 | 1.00 | (Ref) | – |
| >0% to <5% residual activity | Combined | 102 | 23 | 1.38 | 0.89–2.13 | 0.15 |
| | NHM | 85 | 20 | 1.29 | 0.81–2.05 | 0.29 |
| | HM | 17 | 3 | 2.38 | 0.68–8.38 | 0.18 |
| 0% residual activity | Combined | 484 | 121 | 1.53 | 1.21–1.94 | $3.8 \times 10^{-4}$ |
| | NHM | 440 | 117 | 1.52 | 1.19–1.94 | $7.4 \times 10^{-4}$ |
| | HM | 44 | 4 | 1.29 | 0.42–3.90 | 0.66 |

*CRC* colorectal cancer, *NHM* non-hypermutated, *HM* hypermutated, *HR* hazard ratio, *CI* confidence interval.
[a]Cox proportional hazard regression models adjust for age at diagnosis, sex, mutation burden, and study. *TP53* non-silent mutations are based on transcript NM000546.

The presence of a mutation in the RTK/RAS pathway was not significantly associated with CRC survival overall, or when stratified by HM status (Supplementary Data 2). More specifically, the presence of a *BRAF* V600E mutation was not associated with CRC survival, even among cases diagnosed with distant metastatic disease, and the presence of known oncogenic mutations in *KRAS* was only modestly associated with poorer survival (Supplementary Data 2). Although these observations are somewhat contrary to prior studies[10], they are consistent with results based on a large meta-analysis on previously measured tumor marker data including studies that are included here (Phipps et al., in press *Gastroenterology*).

*TGF-beta superfamily pathway*: Members of the TGF-beta superfamily are frequently mutated in sporadic and hereditary CRC causing loss of TGF-beta signaling and its anti-proliferative effects[11]. We sequenced genes encoding for the ligand GDF5, cell membrane-anchored receptors (ACVR2A, ACVR1B, BMPR1A, BMPR2, TGFBR1, and TGFBR2), intracellular receptor-regulated R-Smads (SMAD2, SMAD3), and SMAD4, which is a mediator of signal transduction to the nucleus to regulate expression of target genes. While multiple TGF-beta signaling genes often harbored non-silent mutations in HM tumors, mutated genes appear to be mutually exclusive only in NHM tumors (Supplementary Fig. 3).

Overall, the presence of a mutation in the TGF-beta pathway was associated with less favorable survival for those with NHM CRC (HR = 1.40, 95% CI: 1.06–1.86), but the opposite was true for individuals with HM CRC (HR = 0.44, 95% CI: 0.18–1.06). Among those with NHM tumors, the presence of a somatic mutation in *SMAD4* was most strongly associated with poorer survival (HR = 1.58, 95% CI: 1.13–2.22). However, none of these associations remain significant after adjusting for multiple comparisons.

*IGF2/PI-3-kinase pathway*: In the IGF2/PI-3-kinase pathway, mutually exclusive non-silent mutations were found in the *PIK3CA*, *PIK3R1*, and *PTEN* genes in NHM and HM tumors (Supplementary Fig. 3). 97% of mutations in *PIK3CA*, the catalytic subunit of Phosphoinositide-3-kinase (PI3K), are missense or in-frame indels and 51% of mutations in the regulatory subunit, *PIK3R1*, and 49% of mutations in the negative regulator, *PTEN*, are frameshift, splicing, and truncating mutations. The most mutated gene in this pathway, *PIK3CA*, showed recurrent mutations in codons R88, E542, Q546, H1047. The oncogenic gain of function mutations in codons E542 and H1047 have been described to activate the AKT pathway[12].

The presence of a mutation in the IGF2/PI-3-kinase pathway was not significantly associated with CRC survival overall or when stratified by HM status (Supplementary Data 2).

**MSS-HM tumors without non-silent mutations in *POLE* and *POLD1*.** Approximately 3% of tumors (*n* = 64) were MSS-HM

and without non-silent mutations in *POLE* or *POLD1*. These tumors were frequently mutated in *APC* (73%), *TP53* (50%), and *KRAS* (45%) (Supplementary Fig. 6). The most frequently mutated pathways in these tumors included the WNT-signaling pathway, RTK/RAS signaling pathway, TGF-beta superfamily signaling pathway, and IGF2-PI3K signaling pathway. Slightly more tumors occurred on the right side (53.1%). Across hypermutated tumors, MSS tumors with and without mutations in *POLE/POLD1* and tumors exhibiting MSI showed similar frequency of different types of mutations, such as exonic, intronic, UTR, and intergenic regions (Supplementary Fig. 7).

**Mutated genes by tumor characteristics and sex.** We observed what has been well described by others, that MSI status differs by tumor site and sex, with MSI occurring more frequently in females and in right-sided CRC (Table 2). After accounting for multiple comparisons and adjusting for age, sex, mutational burden, MSI status and study, we observed statistically significant differences in mutation status among right-sided versus left-sided tumors for several genes (Table 2 and Supplementary Data 3), including the *KRAS, TP53, BRAF, BCL9, AMER1,* and *FBXW7,* as well as for several pathways, including RTK/RAS, TP53/ATM, TGF-beta, and IGF2/PI-3-kinase. When stratified by tumor stage, we found that genes mutated in the IGF2/PI3K pathway occurred more frequently in stages 2 and 3 compared to stage 1 tumors (OR = 1.48, *p*-value $7.0 \times 10^{-3}$; Supplementary Data 4). Whereas, genes mutated in the WNT-signaling pathway were less frequent in stages 2 and 3 compared to stage 1 tumors (OR = 0.67, *p*-value $6.6 \times 10^{-3}$). In addition, results were suggestive for an increased frequency of *SMAD4* mutations in stages 2 and 3 tumors compared with stage 1 tumors (OR = 1.91, *p*-value $1.8 \times 10^{-3}$; Supplementary Data 4). Furthermore, mutations in *BRAF* and the RTK/RAS pathway occurred more frequently among females (*BRAF*: OR = 0.37, *p*-value $2.0 \times 10^{-6}$; RTK/RAS pathway mutation: OR = 0.76, *p*-value $2.5 \times 10^{-3}$; Supplementary Data 5).

**Discussion**

In this study, we provide a detailed look at the mutational profile and its link to survival in over 2000 CRC cases. As expected, the most frequently non-silent mutated genes belong to the WNT, TP53/ATM, receptor tyrosine kinase, TGF-beta, and PI-3-kinase pathways; however, mutational frequency varied substantially by hypermutation status. We not only found that HM tumors were associated with improved survival, but also that an increased number of mutations within HM tumors was associated with improved survival. When looking at specific genes while accounting for multiple comparisons, we found that mutations in *TP53* with 0% predicted transcriptional activity were associated with poorer survival. Furthermore, we observed differences in

**Table 2 Distribution of somatic mutated genes and pathways by tumor site.**

| Subtype | Tumor site | | |
|---|---|---|---|
| | Left-sided (n = 1184) | Right-sided (n = 899) | P value[a] |
| MSI status | | | |
| MSI | 47 (4%) | 261 (29%) | 2.77E−07 |
| MSS | 1137 (96%) | 638 (71%) | |
| Hypermutation | | | |
| NHM | 1088 (92%) | 603 (67%) | 0.467 |
| HM | 96 (8%) | 296 (33%) | |
| Mutated genes[b] | | | |
| KRAS[c] | | | |
| Mutated | 379 (32%) | 354 (39%) | 7.79E−10 |
| Non-mutated | 805 (68%) | 545 (61%) | |
| TP53[d] | | | |
| Mutated | 759 (64%) | 381 (42%) | 1.98E−09 |
| Non-mutated | 425 (36%) | 518 (58%) | |
| BRAF[e] | | | |
| Mutated | 23 (2%) | 132 (15%) | 2.89E−05 |
| Non-mutated | 1,161 (98%) | 767 (85%) | |
| BCL9 | | | |
| Mutated | 65 (5%) | 68 (8%) | 4.06E−05 |
| Non-mutated | 1119 (95%) | 831 (92%) | |
| AMER1 | | | |
| Mutated | 57 (5%) | 116 (13%) | 1.10E−04 |
| Non-mutated | 1,127 (95%) | 783 (87%) | |
| FBXW7 | | | |
| Mutated | 174 (14.7%) | 135 (15%) | 1.48E−04 |
| Non-mutated | 1010 (85%) | 764 (85%) | |
| Mutated Pathways[b] | | | |
| RTK/RAS | | | |
| Mutated | 505 (43%) | 586 (65%) | 3.36E−12 |
| Non-mutated | 679 (57%) | 313 (35%) | |
| TP53/ATM | | | |
| Mutated | 789 (67%) | 452 (50%) | 9.62E−08 |
| Non-mutated | 395 (33%) | 447 (50%) | |
| TGF-beta | | | |
| Mutated | 246 (21%) | 422 (47%) | 3.85E−05 |
| Non-mutated | 938 (79%) | 477 (53%) | |
| IGF2/PI-3-kinase | | | |
| Mutated | 172 (15%) | 271 (30%) | 8.23E−05 |
| Non-mutated | 1,012 (85%) | 628 (70%) | |

MSI microsatellite instability, MSS microsatellite stable, HM hypermutated, NHM non-hypermutated.
[a]Analyses adjusted for age at diagnosis, sex, mutation burden, MSI status and study. Significance threshold determined based on Bonferroni correction (205 genes, p value < 2.4 × 10⁻⁴, and 6 pathways, p value < 8.3 × 10⁻³.
[b]Gene and pathway mutation defined based on presence of non-silent mutations in genes.
[c]codons G12, G13, Q61, K117, and A146 mutations.
[d]Transcript NM00546 encoding for the canonical p53 protein was used.
[e]Codon V600 mutations.

mutational frequency of several genes and pathways by tumor location, stage and sex.

As tumors with a large number of mutations are linked to response to immunotherapy due to the larger number of potential neoantigens[13–16], we had a closer look at the HM tumors which account for almost 20% of all CRC. About two-thirds of HM tumors are MSI which occur through defective DNA repair by germline and somatic mutations or promoter methylation in MMR genes that leads to an increased mutational burden. Among the MSS tumors that account for the remaining one-third of HM tumors we observed an enrichment of POLE mutations frequently resulting in an ultra-mutated phenotype. Given our large sample size we were not only able to show that HM tumors were associated with improved survival but further able to show that within

the HM tumors an increasing number of mutations impacted survival positively. This is consistent with the observation that an elevated neoantigen load is associated with high-lymphocytic infiltration and improved CRC-specific survival[7]. In phase II trials, MMR deficient tumors were responsive to the immune checkpoint inhibition using pembrolizumab, nivolumab, and the combination of nivolumab and ipilimumab[13,14,17] which has led to FDA approvals of Opdivo (nivolumab) with or without Yervoy (ipilimumab) and Keytruda (pembrolizumab), for treating tumors with MSI or deficient MMR. While currently such treatment is limited to cases with MSI or MMR deficiency, our data show that these cases only account for about two-thirds of all HM tumors and that nearly one-third of all HM tumors are MSS. These HM-MSS cases may also benefit from checkpoint inhibitor and neoantigen vaccination immunotherapies, particularly as a sizable fraction is even ultra-hypermutated. Accordingly, immunotherapy studies should investigate if the subset of MSS tumors with a large number of mutations would also benefit from this promising treatment. Accordingly, the ongoing clinical trial, NCT01876511 (https://clinicaltrials.gov), extended inclusion to HM-MSS cases. Immunotherapy benefits are also observed in lung cancer with a high tumor mutational burden[18].

Somatic mutations in TP53, which are present in the majority of cancers[3], are associated with poorer clinical outcomes in several cancer types, including CRC[19], consistent with our finding. However, analyses of TP53 somatic mutations and p53 function in relation to cancer outcomes, including CRC, have resulted in inconsistent findings[19–21]. This may be due, in part, to discrepant approaches to defining mutation status, sample sizes, methodologies, and population characteristics. Importantly, when we account for transcriptional activity in the definition of gene mutation status, the relationship between TP53 mutation status and CRC survival became substantially more pronounced. These results demonstrate that while the classification of a mutation as silent or non-silent is relevant for determining the functional effect of a mutation, additional functional annotations may aid in more accurate characterization of the putative effect of mutations on clinical outcomes. Consistent with our finding, a study of CRC and a study of glioma and gastric adenocarcinoma found differences in survival outcomes when accounting for p53 transcriptional activity[22,23] as defined using the IARC TP53 database[9]. Furthermore, a CRC study found that the type of TP53 DNA binding domain mutation affected CRC survival outcome[24]; while using a different functional definition and analyzing a subset of patients, this finding is in line with ours. Mutations in TP53 could result in a gain of oncogenic function, reduced degradation, and dominant-negative effect on the wild-type protein. As such, heterozygous mutations without the loss of heterozygosity could further reduce the activity of p53 protein from wild-type allele by altering the ratios of mutant and wild-type p53 proteins and by generating p53 tetramers with reduced p53 activity. As more information about the functional effects of mutations is gained across other genes, it can be expected that future studies will be able to better characterize mutation phenotypes and clinical impacts of specific somatic mutations. Fortunately, higher throughput functional assays are becoming increasingly available to make such detailed analyses possible.

Our findings suggested that SMAD4 mutations are more frequent in stage 2/3 compared to stage 1 CRC, and are modestly associated with poorer survival; although results were not significant after adjusting for multiple comparisons. In line with this, SMAD4 loss, as determined by immunohistochemistry, was recently reported to be associated with worse CRC survival[25]. The study further showed that SMAD4 loss was associated with resistance to chemotherapy, and decreased tumor immune

infiltration[25]. Large deletions in chromosome 18q were not measured in the present study.

Somatic differences in CRC along the colonic axis from caecum to rectum have been well established. Our findings for differences in MSI, KRAS, BRAF, TP53, and FBXW7 mutations are in line with previous reports of tumor-site differences in these somatic mutations[26,27]. In addition, one study of 516 patients with stage 2 and 3 tumors identified site-specific differences in RTK/RAS, PI-3-kinase, and TGF-beta pathways, which is consistent with our findings, though they did not find statistically significant differences in the mutational frequencies in the TP53/ATM pathway[27]. However, it appears that several studies evaluating gene and pathway mutation frequencies by tumor site did not account for MSI status and mutational burden. Many genes that we identified as having different mutational frequencies by tumor site, while accounting for MSI status and mutational burden, are not as well described in the literature, and additional studies are needed to confirm these. Consistent with our findings, BRAF mutations have been reported to be more prevalent in females[28]. In summary, our findings provide a more detailed understanding of the tumor heterogeneity and how that heterogeneity pertains to prognosis.

In cancers, mutations in CTNNB1 are often found in the hotspot region encoding for codons D32 to S45[2,29,30]. This region contains phosphorylation sites needed for the removal of excess beta-catenin[31]. The beta-catenin degradation complex is formed by APC, AXIN1, and AXIN2, which recruits casein kinase 1 alpha to phosphorylate beta-catenin at S45, and subsequent phosphorylation by GSK3-beta at T41, S37, and S33[32,33]. While S45 phosphorylation priming and subsequent phosphorylation of T41, S37, and S33 are required for the degradation of beta-catenin, the D32 and G34 are also essential for its interaction with BTRCP, a specificity component of ubiquitination machinery[33]. Mutations in this region can stabilize beta-catenin resulting in its cytoplasmic accumulation, nuclear translocation, and transcription of cell cycle- and growth-related genes. Interestingly, we observed that CTNNB1 hotspot mutations were enriched in tumors without inactivating mutations in APC. Treatment of these tumors will require direct inhibition of the beta-catenin/TCF transcriptional complex.

We identified PRKCI, SPZ1, MUTYH, MAP2K4, FETUB, and TGFBR2 as significantly mutated new genes. These genes have been implicated in cancer. Reduced levels of PRKCI correlated with the worst survival in CRC patients, and its deficiency is linked to inflammation and tumorigenesis in mice[34]. SPZ1 promotes epithelial-mesenchymal transition and oncogenesis in liver cancer[35]. This transcription factor functions in the MAPK signaling pathway to stimulate cell proliferation and oncogenesis[36]. MUTYH is involved in oxidative DNA damage repair by removing improperly paired adenine with 8-oxoG[37]. Its deficiency is associated with MUTYH-Associated Polyposis syndrome. As MUTYH is transcriptionally regulated by TP53[38], in tumors with defective TP53, decreased levels of MUTYH could affect DNA repair resulting in accumulation of mutations in cancer genes. The MUTYH mutational signatures persistent with mispaired adenine and 8-oxoG occur frequently in CRC genes including, APC, KRAS, PIK3CA, and SMAD4[37]. Downregulation of MAP2K4, a key mediator of Jun N-terminal kinase signaling, was associated with poor prognosis in CRC patients[39]. FETUB encodes for a cysteine protease inhibitor. Its expression in CRC cell lines and its tumor-suppressing activity in vivo in mice have been described[40]. TGFBR2 is an essential receptor in the TGF-beta signaling pathway, which is often mutated in CRC[2,3,5–7]. Its disruption resulted in invasive intestinal tumors in inflamed mucosa in mice[41]. Our large dataset and a robust mutation calling method have allowed identification of these genes as significantly mutated driver genes.

Independent of activation of EGFR, mutations that activate KRAS, BRAF, or NRAS relay signal to the nucleus to promote cell growth and survival through the RAS-RAF-MAPK pathway. Tumors with these mutations exhibit resistance to anti-EGFR therapy with panitumumab and cetuximab[10,42]. Among the EGF receptor family members, activating mutations in ERBB2 and its dimerization partner, ERBB3, have also been described in CRC[43–45]. A subset of cetuximab-resistant tumors with deregulated ERBB2 pathway was responsive to concurrent treatments with cetuximab and the ERBB2 inhibitor trastuzumab[46]. The oncogenic S310F mutation in ERBB2 responded to inhibition by neratinib, afatinib, and trastuzumab[43]. Treatments of cells harboring V104M mutation in ERBB3 with ERBB antibodies and other inhibitors blocked oncogenic signaling[45]. These two somatic mutations in ERBB2 (S310F, 6% of ERBB2 mutated) and ERBB3 (V104M, 10% of ERBB3 mutated) are found in our CRC cases. These results show that in patients with failed response to EGFR blockade, in addition to KRAS, BRAF, and NRAS sequencing of ERBB2 and ERBB3 is important, as these patients may benefit by ERBB signaling blockade using antibodies or small molecule inhibitors. Furthermore, while most BRAF mutated tumors contain the well-known V600E oncogenic mutation, we also found mutations in neighboring codons D594, L597, and K601 that can lead to resistance to anti-EGFR therapy. Accordingly, extensive mutation profiling of RTK/RAS signaling will allow to identify cases with better treatment options.

The MSS-HM tumors without non-silent mutations in POLE and POLD1 genes appear to have a distinct mutation profile than HM tumors with MSI or POLE/POLD1 mutations. Although most frequently mutated in APC, TP53, and KRAS, resembling MSS-NHM tumors, they are also mutated in other genes with higher frequencies. This subset of tumors comprises 3% of all tumors, and 16% of all HM tumors. These tumors are also likely candidates for immunotherapy like other HM tumors with MSI or mutations in POLE/POLD1 genes.

Our study has some limitations and strengths. We only conducted targeted sequencing on 205 genes. However, these genes were selected based on whole exome sequencing of 1211 CRCs. This large sample allowed us to identify any significantly mutated genes with a variant allele frequency of ≥2.5%[4]. Thus, we expect to have coverage of all important common and even infrequently mutated genes in our panel. While our chosen sequencing technology worked well for DNA from FFPE, we were not able to reliably measure copy number alterations with this technology. Given budget constraints, we were also not able to measure other mutational features, such as epigenetic changes or RNA-Seq, which would have allowed us to more comprehensively capture the various tumor characteristics. Our sample size also limited our ability to examine associations or patterns of alterations in smaller subgroups (e.g., stage strata). However, we were able to conduct the targeted sequencing analysis in over 2000 cases, making this one of the largest studies of somatic mutations in CRC to date. We were able to sequence at a very high coverage, conduct extensive quality control analyses that enable us to account for expected FFPE artifacts and validate mutations using orthogonal methods demonstrating the high quality of our data (for details see "Methods" and Supplementary Methods).

In summary, our study provides insights into the mutational profile of CRC and its potential link to survival. We are providing several findings which lay the foundation to advance better strategies, including personalized approaches for prevention, diagnosis, and treatment for CRC.

## Methods

**Study populations**. We performed targeted sequencing on 2105 cases within GECCO and CCFR. GECCO is an international collaboration to study 130,000

CRC cases from 70 studies from North America, Australia, Asia, and Europe. GECCO focuses on the identification and characterization of genetic risk factors, gene-environment interactions, and the impact of germline genetic, environmental, and lifestyle risk factors on the tumor genome, microbiome, immune response, and survival. The CCFR is a National Cancer Institute-supported consortium consisting of six centers dedicated to the establishment of a comprehensive collaborative infrastructure for interdisciplinary studies in the genetic epidemiology of CRC. The CCFR includes data from ~42,500 total subjects in 15,000 families (10,500 probands, 26,770 unaffected and affected relatives, and 4276 unrelated controls and 923 spouse controls). This study selected tumor samples from two of the CCFR sites, Ontario and Seattle. Study descriptions and sample selection criteria are provided in the Supplementary Text and Supplementary Table 1. Clinical attributes are provided in Supplementary Data 6. The Health Science Research Ethics Board at University of Toronto, Institutional Review Board at Fred Hutch, Mount Sinai Hospital Research Ethics Board, Emory University Institutional Review Board, Research Ethics Board at the Institute of Cancer Research, Ethics Commission Board at Medical University of Vienna, and Ethics Committee of the Medical Faculty of Heidelberg, approved the study, and all patients provided written informed consent to allow the collection of specimens and data used in this analysis.

**Targeted sequencing**. We extracted tumor DNA from FFPE sections and isolated matching normal DNA from the blood, buccal, saliva, or adjacent normal colonic FFPE tissues. Tumor tissue was macrodissected from slides guided by a H&E stained slide marked for the tumor regions. All tumors underwent a pathology review to confirm that the tumor was a primary colorectal carcinoma. We extracted DNA from FFPE tissue using the QIAamp DNA Mini or QIAamp DNA FFPE tissue kits and normal DNA from other tissues using standard DNA extraction methods. DNA concentrations were determined by Quant-iT PicoGreen dsDNA Assay or the Qubit dsDNA HS Assay kits.

DNA extracted from FFPE tissues was subjected to repair by using the PreCR Repair Mix (New England BioLabs, Ipswich, MA). AmpliSeq target amplification was performed using 20 ng of genomic DNA for each of the 2 AmpliSeq primer pools. Following removal of primers, PCR products from each pool were combined and subjected to end repair and A-tailing using the KAPA HyperPrep Kit (Roche). Adapter ligation was performed using the NEXTflex DNA barcodes Kit (PerkinElmer) and libraries were analyzed on High Sensitivity TapeStation and submitted for cluster generation. Barcoded DNA sequence libraries were pooled using 48 samples for tumors and 48 or 192 samples for normal DNA. Paired-end sequencing was performed on HiSeq 2500 using the Illumina Genome Analyzer operating procedure. Paired-end reads were aligned to the reference human genome (GRCh37/hg19) using Burrows-Wheeler Aligner (BWA-MEM version 0.7.9a). Local realignments and base quality recalibrations were performed on aligned data. Only reads aligned uniquely to the reference human GRCh37/hg19 genome assembly were used in downstream analysis.

**Calling somatic variants and significantly mutated genes**. We called somatic single nucleotide variants (SNV) using Strelka v1.0.15[47] and MuTect v1.1.7[48], and took the intersection of mutation calls. We annotated somatic mutation calls by ANNOVAR. A set of additional filters were used such as strand bias, minor allele frequency in Exome Aggregation Consortium (ExAC), read-depth, alternative read-depth, and clustered read position. We further applied an amplicon artifact filtering to remove cases where mutant allele frequency varied across read clusters. We obtained indel calls using majority votes from VarScan2 v2.4.3[49], VarDict (Feb 2017)[50], and Strelka v1.0.15[47]. After initial filtering of indels based on coverage and mutant allele frequency, we noticed some background signals of alternative reads in normal samples. Thus, we used read counts from tumors and normal samples to construct a background filter to remove indel calls in a subset of samples where signals were not significantly higher than background. We used MutSigCV[51] to define significantly mutated genes. For more details on the quality control, calling and analysis see Supplementary methods, Supplementary Fig. 8, and Supplementary Tables 2 and 3.

To define hypermutation status, we plotted point mutations for all samples and observed two very distinct peaks. The minimum value between the two peaks is 23 point mutations per sample (17 mutations per million bases), which we used as a cut-off for defining hypermutation status (Supplementary Fig. 9).

**Validation of point mutations and indels**. We evaluated calls for randomly selected 96 point mutations and 91 indels by Sanger sequencing. Results were used to improve mutation calling accuracy. Subsequently, we conducted a validation study using Sequenom as an orthogonal technology for point mutations and indels. For point mutation calls, we observed false positive and false negative rates of 0.3% and 4.1%, respectively, with a sensitivity of 95.9% and a specificity of 99.7%. As the validation for indels showed room for improvement, we used the data to further tune our calling algorithms. Subsequent Sanger sequencing for another validation of 109 indels showed 93.6% correct calls. In HM-MSS tumors without non-silent mutations in POLE and POLD1 validation of randomly selected mutations (n = 63) by Sanger sequencing showed 95% correct calls.

**MSI status calling**. We called MSI status using mSINGS[52]. Briefly, we established a baseline reference using control samples from peripheral blood and for each of the 169 microsatellite loci included in our panel design, we quantified and compared the number of differently sized repeats in tumor samples to the baseline for the same locus. A locus was considered unstable if the number of mutated alleles exceeded the baseline reference by three times the standard deviation. To define an MSI positive tumor, we evaluated the fraction of unstable microsatellite loci out of the total number of loci analyzed as well as a qualitative separation of samples (a cutoff fraction of 10% unstable loci, Supplementary Fig. 10). Of 2105 tumor samples, we assigned 310 MSI positive with a mean fraction of 0.27 unstable loci (range = 0.13–0.45; SD = 0.066) and 1795 MSI-negative with a mean fraction of 0.04 unstable loci (range = 0.01–0.12; SD = 0.019). As an additional way to validate calls, we compared classification of MSI status results for participants that had both existing tumor marker data and determined tumor characteristics from the targeted sequencing data. The classifications from orthogonal approaches were highly concordant with 98.6% concordance for the 1534 individuals with information on MSI status from both sources.

**Definition of mutated gene and pathways**. *Mutations*: We defined gene mutations based on the presence of non-silent mutations as determined by ANNO-VAR[53] refGene annotations. A SNV was considered to be non-silent if it was annotated as exonic and nonsynonymous, stop-gain, stop-loss, or splicing. An indel was considered to be non-silent if it was annotated as exonic and a frameshift deletion, frameshift insertion, in-frame deletion, in-frame insertion, stop-gain, or stop-loss. For a subset of genes, we further refined definitions based on known annotations regarding functional effects of mutations (e.g., V600E mutation in BRAF).

*Pathways*: We defined primary pathways implicated in CRC[2] for downstream analyses (see Supplementary Fig. 3 for list of genes included in each pathway). A pathway was considered mutated if any gene within that pathway had a non-silent mutation.

*TP53 residual activity prediction*: TP53 residual activity was determined using the IARC TP53 database (version R19, August 2018)[9,22]. TP53 non-silent SNVs were classified into subgroups with 0, 0–5%, and >5% residual transcriptional activities. Tumors with truncating mutations in TP53 and without TP53 mutations were classified into subgroups with 0 and >5% residual activities, respectively.

**Statistical analyses**. We used Bonferroni corrected p-values to assess statistical significance, accounting for the number of genes or pathways tested in each type of analysis.

*Survival analyses*: We used Cox proportional hazards regression to estimate adjusted HRs and 95% CIs for the association of mutated genes and pathways with CRC-specific survival. Person time accrued from the date of diagnosis to the date of death or the end of follow-up. Cases who died from causes other than CRC were censored at the date of death. We examined proportional hazards assumptions by testing for a nonzero slope of the scaled Schoenfeld residuals as a function of survival time. All analyses were adjusted for age at diagnosis, sex, and study. Primary analyses were conducted without adjustment for hypermutation status, to allow for the possibility of effect modification by this attribute; however, in instances in which associations were observed to be similar across HM and NHM cases, we conducted additional analyses adjusting for hypermutation status. Primary analyses of survival were also not adjusted for stage at diagnosis, to account for the fact that meaningful associations between somatic mutations and survival could operate via an impact on disease aggressiveness (and therefore stage). In instances in which we observed significant differences in the prevalence of mutated genes or pathways by stage at diagnosis or tumor site, we also carried out analyses stratified by and adjusted for these attributes.

*Case only tumor comparison*: We used a $\chi^2$ test to compare gene and pathway mutation frequencies in HM and NHM tumors. To evaluate potential differences in mutated genes and pathways by tumor site, tumor stage, and sex, we ran unconditional case-only logistic regression analyses adjusting for MSI status. We defined left-sided tumors as those occurring in the splenic flexure, descending colon, sigmoid colon, rectosigmoid junction, and rectum, and right-sided tumor as transverse colon, hepatic flexure, ascending colon, and cecum. We excluded appendix and anal cancer cases from all analyses. We did not stratify further by rectum as the TCGA analysis did not observe substantial differences between colon vs rectum[2]. In analyses comparing mutation frequency by tumor site, we used left-sided tumors as the referent category. Stage 1 cases served as the referent category in analyses of mutation frequency by stage at diagnosis. In analyses comparing mutation frequency by sex, we used females as the referent category.

**Reporting summary**. Further information on research design is available in the Nature Research Reporting Summary linked to this article.

## Data availability

All data generated or analyzed during this study are included in this published article (and its supplementary information files). The original sequencing data are available at the database of Genotypes and Phenotypes (dbGaP, accession phs002050.v1.p1). IARC TP53 data are available at https://p53.iarc.fr.

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

## Acknowledgements

Fred Hutch core grant: This research was funded in part through the NIH/NCI Cancer Center Support Grant P30 CA015704. GECCO: National Cancer Institute, National Institutes of Health, U.S. Department of Health and Human Services (U01 CA137088, U01 CA164930 and R01 CA176272). CCFR: The Colon Cancer Family Registry (CFR) is supported by funding from the National Cancer Institute (NCI), National Institutes of Health (NIH) (grant numbers U01 CA167551). The Colon CFR participant recruitment and collection of data and biospecimens used in this study were supported through cooperative agreements with Ontario Familial Colorectal Cancer Registry (NCI/NIH U01/U24 CA074783) and Seattle Colorectal Cancer Family Registry (NCI/NIH U01/U24 CA074794). The content of this manuscript does not necessarily reflect the views or

policies of the National Cancer Institute or any of the collaborating centers in the Colon Cancer Family Registry (CCFR), nor does mention of trade names, commercial products, or organizations imply endorsement by the US Government, any cancer registry, or the CCFR. PMH study is funded by National Institutes of Health (R01 CA076366 to P.A.N). CPS-II: The American Cancer Society funds the creation, maintenance, and updating of the Cancer Prevention Study-II (CPS-II) cohort. This study was conducted with Institutional Review Board approval. DACHS: This work was supported by the German Research Council (BR 1704/6-1, BR 1704/6-3, BR 1704/6-4, CH 117/1-1, HO 5117/2-1, HE 5998/2-1, KL 2354/3-1, RO 2270/8-1 and BR 1704/17-1), the Interdisciplinary Research Program of the National Center for Tumor Diseases (NCT), Germany, and the German Federal Ministry of Education and Research (01KH0404, 01ER0814, 01ER0815, 01ER1505A and 01ER1505B). OFCCR: National Institutes of Health, through funding allocated to the Ontario Registry for Studies of Familial Colorectal Cancer (U01 CA074783); see CCFR section above. Additional funding toward genetic analyses of OFCCR includes the Ontario Research Fund, the Canadian Institutes of Health Research, and the Ontario Institute for Cancer Research, through generous support from the Ontario Ministry of Research and Innovation. CORSA: We thank all those who agreed to participate in the CORSA study, including the patients and the control persons, as well as all the physicians and students. CPS-II: The authors thank the CPS-II participants and Study Management Group for their invaluable contributions to this research. The authors would also like to acknowledge the contribution to this study from central cancer registries supported through the Centers for Disease Control and Prevention National Program of Cancer Registries, and cancer registries supported by the National Cancer Institute Surveillance Epidemiology and End Results program. DACHS: We thank all participants and cooperating clinicians, and Ute Handte-Daub, Utz Benscheid, Muhabbet Celik and Ursula Eilber for excellent technical assistance. GECCO: The authors would like to thank all those at the GECCO Coordinating Center for helping bring together the data and people that made this project possible. PMH: The authors would like to thank the study participants and staff of the Hormones and Colon Cancer study. Ontario Institute for Cancer Research (OICR): The authors would like to thank Faridah Mbabaali, Carolina Bocanegra, and Tanya Mohanta for assistance in sequencing. We thank Dr. Robert Campos, Head, Research Operations, Interim Director, Genomics Technology Program, for supporting sequencing at OICR. OICR is supported by the Ontario Ministry of Research and Innovation, Ontario, Canada. Genome Quebec Innovation Centre, Montreal, Canada, for conducting Sequenom assays; and Priscilla Hunt from Agena Bioscience, San Diego, CA, for assistance in designing the Sequenom assays.

## Author contributions

S.H.Z., T.A.H., A.I.P., R.S., S.I.B., S.R.H., E.R.M., J.D.M., N.P., S. Gallinger, T.J.H., and U.P. conceived and designed the experiments. S.H.Z., T.A.H., A.J.F., P.M.K., J.D.M, J.K.M., D.P., A.R., E.E.R., C.D.T., L.T., M.V.T., S.N.T., Q.M.T., T.J.H., and U.P. performed the experiments. S.H.Z., T.A.H., A.I.P., R.S., Q.M.T., B.L.B., C.S.F., L.A.G., P.G., M. Giannakis, C.S.G., J.B.A., R.T.B., I.B., C.M.C., L.E.H., A.H., J.R.H., M.L., Y.L., X.L., X.J.M., R.N., C.Q., M.J.Q., E.S., B.H.S., L.D.S., C.Y.U., X.W., D.A.W., C.K.Y., W.S., L.H., and U.P. analyzed the data.

S.H.Z., T.A.H., A.I.P., R.S., Q.M.T., B.L.B., P.G., M. Giannakis, E.A., E.L.A., H.B., S.B., D.D.B., Y.C., A.T.C., J.C-C., D.A.D., A.B.F., J.C.F., C.S.F., L.A.G., S. Gruber, M. Guinter, S.H., J.L.H., W-Y.H., V.M., X.J.M., E.S., L.D.S., B.V.G., D.A.W., S.O., A.G., P.A.N., M.A.J., S. Gallinger, M.H., P.T.C., S.N.T., W.S., T.J.H. and U.P. contributed reagents, materials, and analysis tools. H.B., D.D.B., A.T.C., J.C-C., J.C.F., C.S.F., S. Gruber, J.L.H., W-Y.H., V.M., B.V.G., S.O., A.G., P.A.N., S. Gallinger, M.H., P.T.C., and S.N.T. designed and executed one of the studies included in the effort. S.H.Z., T.A.H., A.I.P. T.J.H., and U.P. wrote the first draft. All authors reviewed the manuscript and provided intellectual content.

## Competing interests

D.B. served as consultants on the Tumour Agnostic (dMMR) Advisory Board of Merck Sharp and Dohme in 2017 and 2018 for Pembrolizumab. E.M. received stocks and honoraria, from Qiagen N.V. as a member of the supervisory board, and from PACT Pharma LLC as a member of the scientific advisory board. X.M., L.G., and E.S. are currently employed by Pfizer, Roche, and NeoGenomics, respectively. All other authors declare no competing interests.

## Additional information

Syed H. Zaidi[1,51], Tabitha A. Harrison[2,51], Amanda I. Phipps[2,51], Robert Steinfelder[2], Quang M. Trinh[1], Conghui Qu[2], Barbara L. Banbury[2], Peter Georgeson[3,4], Catherine S. Grasso[2,5], Marios Giannakis[6,7], Jeremy B. Adams[1], Elizabeth Alwers[8], Efrat L. Amitay[8], Richard T. Barfield[2,9], Sonja I. Berndt[10], Ivan Borozan[1], Hermann Brenner[8,11,12], Stefanie Brezina[13], Daniel D. Buchanan[3,4,14,15], Yin Cao[16,17,18], Andrew T. Chan[7,16,17,19,20,21], Jenny Chang-Claude[22,23], Charles M. Connolly[2], David A. Drew[16,17], Alton Brad Farris III[24], Jane C. Figueiredo[25,26], Amy J. French[27], Charles S. Fuchs[28,29,30], Levi A. Garraway[6,7], Steve Gruber[31], Mark A. Guinter[32], Stanley R. Hamilton[33], Sophia Harlid[34], Lawrence E. Heisler[1], Akihisa Hidaka[2], John L. Hopper[4], Wen-Yi Huang[10], Jeroen R. Huyghe[2], Mark A. Jenkins[35], Paul M. Krzyzanowski[1], Mathieu Lemire[1], Yi Lin[2], Xuemei Luo[1], Elaine R. Mardis[36], John D. McPherson[1,37], Jessica K. Miller[1], Victor Moreno[38,39,40], Xinmeng Jasmine Mu[6,7], Reiko Nishihara[41,42,43], Nickolas Papadopoulos[44], Danielle Pasternack[1], Michael J. Quist[2], Adilya Rafikova[1], Emma E. G. Reid[1], Eve Shinbrot[45], Brian H. Shirts[46], Lincoln D. Stein[1], Cherie D. Teney[1], Lee Timms[1], Caroline Y. Um[32],

Bethany Van Guelpen [34,47], Megan Van Tassel[1], Xiaolong Wang[2], David A. Wheeler [45], Christina K. Yung [1], Li Hsu[2,48], Shuji Ogino[7,20,41,43], Andrea Gsur [13], Polly A. Newcomb [2,9], Steven Gallinger[1,49,50], Michael Hoffmeister[8], Peter T. Campbell[32], Stephen N. Thibodeau[27], Wei Sun[2], Thomas J. Hudson[1] & Ulrike Peters [2,9✉]

[1]Ontario Institute for Cancer Research, 661 University Avenue, Toronto, ON M5G 0A3, Canada. [2]Public Health Sciences Division, Fred Hutchinson Cancer Research Centre, 1100 Fairview Ave N, Seattle, WA 98109, USA. [3]Colorectal Oncogenomics Group, Department of Clinical Pathology, The University of Melbourne, Parkville, Victoria, Australia. [4]University of Melbourne Centre for Cancer Research, Victorian Comprehensive Cancer Centre, Parkville, VIC 3010, Australia. [5]Department of Surgery, Cedar Sinai Medical Center, 8700 Beverly Blvd, Los Angeles, CA 90048, USA. [6]Department of Medical Oncology, Dana-Farber Cancer Institute and Harvard Medical School, 450 Brookline Avenue, Boston, MA 02215, USA. [7]Broad Institute of MIT and Harvard, 415 Main St, Cambridge, MA 02142, USA. [8]Division of Clinical Epidemiology and Aging Research, German Cancer Research Center (DKFZ), Im Neuenheimer Feld 280, 69120 Heidelberg, Germany. [9]Department of Epidemiology, University of Washington, Seattle, WA, USA. [10]Division of Cancer Epidemiology and Genetics, National Cancer Institute, National Institutes of Health, Bethesda, MD 20892, USA. [11]Division of Preventive Oncology, German Cancer Research Center (DKFZ) and National Center for Tumor Diseases (NCT), Heidelberg, Germany. [12]German Cancer Consortium (DKTK), German Cancer Research Center (DKFZ), Heidelberg, Germany. [13]Institute of Cancer Research, Department of Medicine I, Medical University of Vienna, Borschekagasse 8a, 1090 Vienna, Austria. [14]Genomic Medicine and Family Cancer Clinic, The Royal Melbourne Hospital, Parkville, VIC, Australia. [15]Centre for Epidemiology and Biostatistics, Melbourne School of Population and Global Health, The University of Melbourne, Melbourne, Australia. [16]Clinical and Translational Epidemiology Unit, Massachusetts General Hospital and Harvard Medical School, 100 Cambridge Street, Boston, MA 02114, USA. [17]Division of Gastroenterology, Massachusetts General Hospital and Harvard Medical School, 25 Shattuck Street, Boston, MA 02115, USA. [18]Division of Public Health Sciences, Washington University School of Medicine in St. Louis, St. Louis, MO, USA. [19]Channing Division of Network Medicine, Brigham and Women's Hospital and Harvard Medical School, Boston, MA, USA. [20]Department of Epidemiology, Harvard T.H. Chan School of Public Health, Harvard University, 677 Huntington Avenue, Boston, MA 02115, USA. [21]Department of Immunology and Infectious Diseases, Harvard T.H. Chan School of Public Health, Harvard University, 677 Huntington Avenue, Boston, MA 02115, USA. [22]Division of Cancer Epidemiology, German Cancer Research Center (DKFZ), Im Neuenheimer Feld 280, 69120 Heidelberg, Germany. [23]Cancer Epidemiology Group, University Medical Centre Hamburg-Eppendorf, University Cancer Centre Hamburg (UCCH), 20246 Hamburg, Germany. [24]Pathology and Laboratory Medicine, Emory University School of Medicine, 1364 Clifton Rd. NE, Atlanta, GA 30322, USA. [25]Department of Medicine, Samuel Oschin Comprehensive Cancer Institute, Cedars-Sinai Medical Center, 8700 Beverly Blvd, Los Angeles, CA 90048, USA. [26]Department of Preventive Medicine, Keck School of Medicine, University of Southern California, Los Angeles, CA, USA. [27]Division of Laboratory Genetics, Department of Laboratory Medicine and Pathology, Mayo Clinic, 200 First Street SW, Rochester, MN 55905, USA. [28]Yale Cancer Center, 333 Cedar St, New Haven, CT 06510, USA. [29]Department of Medicine, Yale School of Medicine, New Haven, CT, USA. [30]Smilow Cancer Hospital, New Haven, CT, USA. [31]Department of Preventive Medicine, USC Norris Comprehensive Cancer Center, Keck School of Medicine, University of Southern California, 1441 Eastlake Avenue, Los Angeles, CA 90089, USA. [32]Behavioral and Epidemiology Research Group, American Cancer Society, 250 Williams St NW, Atlanta, GA 30303, USA. [33]City of Hope Comprehensive Cancer Center, 1500 East Duarte Road, Duarte, CA 91010, USA. [34]Department of Radiation Sciences, Oncology Unit, Umeå University, SE-901 87 Umeå, Sweden. [35]Centre for Epidemiology and Biostatistics, Melbourne School of Population and Global Health, The University of Melbourne, Melbourne, VIC 3010, Australia. [36]Institute for Genomic Medicine, Nationwide Children's Hospital and The Ohio State University College of Medicine, 575 Children's Crossroad, Columbus, OH 43215, USA. [37]Department of Biochemistry and Molecular Medicine, UC Davis School of Medicine, 2700 Stockton Blvd, Sacramento, CA 95817, USA. [38]Oncology Data Analytics Program, Catalan Institute of Oncology and ONCOBELL Program, IDIBELL, Avinguda de la Granvia de l'Hospitalet 199-203, 08908, L'Hospitalet de Llobregat, Barcelona, Spain. [39]CIBER Epidemiología y Salud Pública (CIBERESP), 28029 Madrid, Spain. [40]Department of Clinical Sciences, Faculty of Medicine, University of Barcelona, 08907 Barcelona, Spain. [41]Department of Oncologic Pathology, Dana-Farber Cancer Institute and Harvard Medical School, 450 Brookline Avenue, Boston, MA 02215, USA. [42]Department of Nutrition, Harvard T.H. Chan School of Public Health, Boston, MA, USA. [43]Program in MPE Molecular Pathological Epidemiology, Department of Pathology, Brigham and Women's Hospital and Harvard Medical School, 75 Francis Street, Boston, MA 02115, USA. [44]Department of Oncology, Sidney Kimmel Comprehensive Cancer Center, The Johns Hopkins Institution, 1650 Orleans Street, Baltimore, MD 21231, USA. [45]Human Genome Sequencing Centre, Baylor College of Medicine, One Baylor Plaza, Alkek N1419, Houston, TX 77030, USA. [46]Department of Laboratory Medicine, University of Washington, University of Washington Medical Center, 1959 NE Pacific St, Seattle, WA 98195, USA. [47]Wallenberg Centre for Molecular Medicine, Umeå University, SE-901 87 Umeå, Sweden. [48]Department of Biostatistics, University of Washington, Seattle, WA, USA. [49]Lunenfeld Tanenbaum Research Institute, Mount Sinai Hospital, University of Toronto, Toronto, ON, Canada. [50]Department of Surgery, University Health Network Toronto General Hospital, Toronto, ON, Canada. [51]These authors contributed equally: Syed H. Zaidi, Tabitha A. Harrison, Amanda I. Phipps. ✉email: upeters@fredhutch.org

