## [Peer Review File · Nature Communications]

Reviewers' comments:

Reviewer #1 (Remarks to the Author):

In this study, Zaidi et al. performed deep targeted sequencing using a custom 205 gene AmpliSeq assay, including MSI status, in 2105 CRC cases with matched normals. The clinical annotation of the cases includes disease-specific survival data. Their main findings are the discovery of novel genes mutated in CRC, including PRKCI, SPZ1, MUTYH, MAP2K4, FETUB, TGFB2. Within hypermutated tumors, they found a correlation between mutation rate and outcome. They also found TP53 mutations to be associated with the poorest prognosis.

While the data set certainly has potential, there are several major issues with the manuscript. The analyses generally follow the pattern of the TCGA manuscript without too much novelty. Furthermore, because the authors did not include the variant calls, their findings are currently not reproducible and a careful review of the data therefore not possible. In the absence of the mutation data, it appears that there are some issues with the mutation data that need to be corrected.

The authors should address the following aspects to improve the study:

1. The authors state that the data will be made available via dbGaP. In addition, however, the variant calls as well as all clinical attributes used in the analysis should be made available in the supplemental material and/or through a variant archive. For example, the data set could be deposited into the cBioPortal for Cancer Genomics.
2. The authors find a rate of 19% for hypermutated tumors. This is much higher than reported in other studies (8.3% in TCGA, 8.7% in Yaeger et al). The overall mutation rate of the cohort was also higher than reported in previous studies (14.24 mutations per sample in 205 genes, vs 12.85 mutations per gene in Yaeger et al., a study that sequenced more than twice the number of genes), suggesting potential issues with the mutation calls. To make the data better comparable to other data sets, the mutation rates should be reported as mutations per megabase.
3. The authors found that a relatively large fraction (nearly one third) of hypermutated tumors were neither MSI-H nor had a POLE mutation. This number is surprising but could be interesting if real. Running MSIsensor, or a similar method, as an additional method to validate this finding could be helpful. This finding may, however, also be related to the apparent issue of an increased mutation count.
4. The outcome analysis, which found BMP2, ZFH3, KMT2B, and RYR1 to be associated with better outcome is biased by the fact that these genes are more commonly mutated in hypermutated cases. While the authors acknowledge this ("In analyses stratified by hypermutated status, no genes were significantly associated with CRC-specific survival after accounting for multiple comparisons"), the presentation of the finding about these 4 genes is misleading and should be under-emphasized.
5. In the pathway analysis, the authors should restrict analysis to only known or likely driver variants. They currently consider all variants, which, especially in hypermutated tumors, includes many variants of unknown significance.
6. Identifying statistically recurrently mutated genes in hypermutated tumors is notoriously difficult, so the claim to have found novel genes is rather weak. Furthermore, genes such as TGFB2 or BMP2 harbor homopolymer stretches, which are often affected by frameshift insertions or deletions in MSI cases.
7. The authors describe CTNNB1 mutations in Figure 4. Did the study identify the exon 4 in-frame deletions as well that were recently described in colorectal cancer? These mutations had previously

been missed in other studies due to the mutation callers used.

8. APC Mutation frequency in non hypermutated cases (62.7%) appears to be low compared to TCGA (81%) and Yaeger et al. (79%).

9. The regression model used to evaluate differences in mutation status between tumor stage, tumor site, and sex lacks clarity, as it is unclear how these variables were corrected for in the model.

10. Figure 3 does not convey any novel information.

Reviewer #2 (Remarks to the Author):

Zaidi et al. perform targeted sequencing for a panel of 205 colorectal cancer-associated genes in >2000 colorectal tumours to identify a number of recurrently mutated genes that had not met significance thresholds and to identify a number of genes that confer a favourable prognosis when mutated. They also show that approximately 1/3 hypermutated tumours that ought to benefit from checkpoint inhibitors do not have detectable micro satellite instability or POLE mutations and so might be missed by existing clinical assays. Previous studies such as MSK-IMPACT had instead termed any hypermutated samples without POLE mutations as 'MSI-H/hypermutated', which was likely misleading.

This paper should be viewed in context of the recent MSK-IMPACT study which performed amplicon-based sequencing on 1134 mostly metastatic colorectal tumours. They were able to determine mutations, copy-number alterations and DNA-level rearrangements for a subset of genes.

The main differences between Zaidi et al.'s study and MSK-IMPACT is that a) cohort size is larger thus providing enhanced statistical power to identify recurrent mutations and b) Zaidi et al. only report SNVs and indels, c) Zaidi et al. superimpose survival data (although similar data has been reported by the TCGA project).

To my mind the authors have not sufficiently followed-up or given due attention to the novel findings of this project and have instead focussed too much on results that have been previously described in substantial detail by TCGA/MSK-IMPACT (e.g. improved survival in HM tumours, patterns of pathway activation).

There is a need to expand on:

Biological relevance - recurrent mutations in PRKCI, SPZ1, MUTYH, MAP2K4 (e.g. component of MAPK pathway, mutants cells shown to be sensitive to MEK inhibitors), FETUB

Translational relevance - survival association with BMP2, ZFX3, KMT2B, and RYR1; proportion of hypermutated tumours probably missed by existing assays (e.g. speculate on how these tumours must be identified in a cost-effective way)

Specific comments

There are a number of strengths to the study including the size of the cohort, the robust mutation calling strategy using consensus calls from more than one mutation calling algorithm, successful validation of a random subset of mutation calls using Sanger sequencing and well justified selection of hypermutated samples

The major weakness with the study is the lack of copy number alteration data which is a significant problem - recent work has shown that it is predominantly CNV that drives precursor lesion

transformation and subclonal evolution in cancers (Cross et al, 2018). Indeed the title of the paper is thus somewhat misleading as it is only the SNV somatic mutation landscape that is described in this paper.

This is a particular problem in 2 sections:

1. Page 11 the section ascribing the residual transcriptional activity of p53, as only 5% of mutations carried biallelic SNV's. Therefore a large assumption is made that the single allele mutations were associated with LOH in the other allele. This doesn't appear to have been shown.

2. Page 15 - the section discussing the favourable prognostic implication of BMPR2 (quoting Voorneveld et al 2014). The latter paper was very clear that the role of BMP signalling is skewed to pro-tumorigenic following loss of SMAD4. Although SNV mutation of SMAD4 is measured, this doesn't account for chrom 18q loss which is the most common way of impacting this pathway in CRC.

Other comments.

It would be interesting to do further analysis (?CNV, ?methylation) on the 40 or so tumours with very low SNV mutation burden (<5 mutations). What is driving these lesions?

The hyper and ultramutated tumours without evidence for POLE or MMR are interesting and clinically relevant, as discussed in the manuscript. However the reader is left somewhat hanging as to the possible cause of this molecular phenotype. Were other replication fork genes in the targeted panel? Should these tumours undergo more extensive omic analysis to look for novel drivers of mutator phenotype?

There is lack of clinicopathological information regarding tumours - age, sample histology e.g. mucinous/signet ring, confirmation that all samples were reviewed by expert histopathological prior to extraction, length of follow-up available for determination of survival data

Some of the 'novel' recurrent mutations have been previously described - TGFBR2 (<https://www.ncbi.nlm.nih.gov/pubmed/17985359>)

Re: our manuscript "*Landscape of somatic mutations in colorectal cancer and the impact on survival*, NCOMMS-19-18803-A-Z"

Responses to Reviewers' comments

Reviewer #1 (Remarks to the Author):

In this study, Zaidi et al. performed deep targeted sequencing using a custom 205 gene AmpliSeq assay, including MSI status, in 2105 CRC cases with matched normals. The clinical annotation of the cases includes disease-specific survival data. Their main findings are the discovery of novel genes mutated in CRC, including PRKCI, SPZ1, MUTYH, MAP2K4, FETUB, TGFBR2. Within hyper-mutated tumors, they found a correlation between mutation rate and outcome. They also found TP53 mutations to be associated with the poorest prognosis.

While the data set certainly has potential, there are several major issues with the manuscript. The analyses generally follow the pattern of the TCGA manuscript without too much novelty. Furthermore, because the authors did not include the variant calls, their findings are currently not reproducible and a careful review of the data therefore not possible. In the absence of the mutation data, it appears that there are some issues with the mutation data that need to be corrected.

Given that TCGA was one of the first comprehensive analysis into the somatic mutational landscape of colorectal cancer, a relatively small number of similar studies have been conducted since then and genes included in our panel were selected from the TCGA data, we believe that a consistent analysis with TCGA is an important starting point to show consistencies and differences. As our study includes almost 10 times the sample size of the first TCGA paper (Nature 2012), our study can provide further insights given improved statistical power. We also like to note that we have information on CRC-specific survival, while TCGA has only data on overall survival, which tends to weaken associations between mutated genes and survival. As such finding for survival and stage (or the lack of) in this sizable dataset are particularly important for the design of future studies.

We agree that providing information on the variant calls is important and we now include these in Supplementary Table 1.

The authors should address the following aspects to improve the study:

1. The authors state that the data will be made available via dbGaP. In addition, however, the variant calls as well as all clinical attributes used in the analysis should be made available in the supplemental material and/or through a variant archive. For example, the data set could be deposited into the cBioPortal for Cancer Genomics.

Variant calls and clinical attributes are now included in Supplementary Tables 1 and 7. Please note that dbGaP is the only approved public database that was approved by ethics committees. dbGAP is an NIH approved database for submission of sensitive information involving personal health information and genotype data.

2. The authors find a rate of 19% for hypermutated tumors. This is much higher than reported in other studies (8.3% in TCGA, 8.7% in Yaeger et al). The overall mutation rate of the cohort was also higher than reported in previous studies (14.24 mutations per sample in 205 genes, vs 12.85 mutations per gene in Yaeger et al., a study that sequenced more than twice the number of genes), suggesting potential issues with the mutation calls. To make the data better

comparable to other data sets, the mutation rates should be reported as mutations per megabase.

We note that in the TCGA publication (TCGA, Nature 2012), 16% of the 276 tumors (not 8.3%) were found hypermutated, . In a large study conducted at the Dana Farber Cancer Institute, the sequencing of 619 colorectal tumors identified 21% hypermutated tumors (Giannakis et al 2016). Our finding of 19% hypermutated is in line with these publications.

The study by Yeager et al (Cancer Cell, 2018) is focussed on metastatic CRC. As hypermutated tumors are less likely to develop metastasis, it is expected that the fraction of hypermutated tumors is lower in the study by Yeager et al. Furthermore, the estimate of 8.7% only includes MSI-high tumors. As we have shown about 1/3 of the hypermutated tumors are MSS.

In our study, the mutation rate is slightly higher than Yeager et al. This probably can be because our panel included only genes prioritized for higher mutation frequencies in CRC while Yeager et al used a pan-cancer panel including genes selected for other cancer sites. In Yeager's dataset at cBioPortal, there are 16.8 mutations per sample in 601 primary tumors and 8.8 mutations per sample in 533 metastatic tumors. Combined analysis of primary and metastatic tumors, gives 13 mutations per sample. There are major differences between the two studies, including the number of genes sequenced, sample types, and inclusion of intronic or exonic mutations, which does not allow an accurate comparison of mutation rates between the two studies. Furthermore, Yeager included a smaller fraction of hypermutated tumors which contributes to the lower mutation rate. As suggested by the Reviewer, we have included mutations per megabases in the text.

Please note that the Reviewer 2 found a robust mutation calling strategy, successful validation of a random subset of mutation calls, and well-justified selection of hypermutated samples, as the main strengths of our manuscript. For point mutation calls, we observed false positive and false negative rates of 0.3% and 4.1%, respectively, with a sensitivity of 95.9% and a specificity of 99.7%. Moreover, validation of indels by Sanger sequencing showed 93.6% correct calls.

3. The authors found that a relatively large fraction (nearly one third) of hypermutated tumors were neither MSI-H nor had a POLE mutation. This number is surprising but could be interesting if real. Running MSIsensor, or a similar method, as an additional method to validate this finding could be helpful. This finding may, however, also be related to the apparent issue of an increased mutation count.

The Reviewer suggested an additional method to validate the MSI data. Before analyzing data by mSINGS, we evaluated its performance against available previously-derived MSI calls of the same tumors that had been generated using established methods (i.e., genomic testing using the Bethesda Guidelines or immunohistochemistry for mismatch repair proteins). We found high concordance between historic MSI data and the MSI calls generated by mSINGS. This information is now included in the Methods section.

4. The outcome analysis, which found BMP2, ZFH3, KMT2B, and RYR1 to be associated with better outcome is biased by the fact that these genes are more commonly mutated in hypermutated cases. While the authors acknowledge this ("In analyses stratified by hypermutated status, no genes were significantly associated with CRC-specific survival after accounting for multiple comparisons"), the presentation of the finding about these 4 genes is misleading and should be under-emphasized.

We have revised our text to make the presentation of these findings more transparent. Given that analyses stratified by hypermutated status are subject to much smaller numbers (and, therefore, lower power), we feel that it is worth mentioning those instances in which we observed significant findings in analyses of all cases. However, we appreciate the reviewer's comment that our presentation was misleading, and have rearranged the presentation to provide stronger emphasis to the fact that analyses stratified by hypermutated status were not statistically significant.

5. In the pathway analysis, the authors should restrict analysis to only known or likely driver variants. They currently consider all variants, which, especially in hypermutated tumors, includes many variants of unknown significance.

Consistent with previous publications, we restricted description and analyses of single genes to coding non-synonymous variants. We believe it is best to be consistent in our analytical approach for both single genes and pathway analysis. We also note that unfortunately, for most genes there is no well-defined list of putative driver variants. However, in cases where we had this information, we also conducted single gene analysis restricted to these putative driver mutations. As we have shown in the case of *TP53*, including information on functional activity improved the signal for survival. Unfortunately, there are far too few genes for which we can define putative driver mutations to allow a pathway analysis. Furthermore, in a large dataset, there is an increased possibility to discover new rare driver mutations. Excluding these would underestimate the impact of these mutated genes on CRC.

6. Identifying statistically recurrently mutated genes in hypermutated tumors is notoriously difficult, so the claim to have found novel genes is rather weak. Furthermore, genes such as *TGFBR2* or *BMP2R* harbor homopolymer stretches, which are often affected by frameshift insertions or deletions in MSI cases.

We agree with the reviewer that identifying statistically recurrently mutated genes in hypermutated tumors is difficult. Please note that we also performed Sanger sequencing validation of 77 of the mutations from the newly identified significantly mutated genes, *PRKC1*, *SPZ1*, *MUTYH*, *MAP2K4*, *FETUB*, and *TGFBR2*, showing a concordance of 98.7%. We have now included this information in the Results. We also found that 25 of the 99 tumors mutated in *TGFBR2* in our study carry a frameshift deletion in tumors with MSI. This mutation is present in the TCGA and absent in the datasets from Giannakis et al and Yaeger et al. It is plausible that due to the absence of this common mutation in these studies, *TGFBR2* was not previously identified as a significantly mutated gene. We also have added details on the potential link of each gene to CRC, which further supports our findings.

7. The authors describe *CTNNB1* mutations in Figure 4. Did the study identify the exon 4 in-frame deletions as well that were recently described in colorectal cancer? These mutations had previously been missed in other studies due to the mutation callers used.

The recently described large in-frame deletions in *CTNNB1* were identified by targeted sequencing included capture probes for the intronic regions. Our AmpliSeq panel only includes coding exons and flanking exon-intron junctions. Identification of large deletions spanning introns is not possible with the current AmpliSeq amplicon-based panel. However, we have identified in-frame deletions of S45 in the *CTNNB1* gene. This deletion is present in TCGA but not in the recent study describing large deletions spanning introns. Accordingly, we see these findings as complementary.

8. APC Mutation frequency in non hypermutated cases (62.7%) appears to be low compared to TCGA (81%) and Yaeger et al. (79%).

Please note that we only included non-synonymous coding mutations to estimate 62.7% of the non-hypermutated primary tumors with *APC* mutations. Giannakis et al 2016 also reported *APC* mutations in 62% of non-hypermutated tumors. Our finding is similar to the sequencing study of 619 primary CRC tumors by Giannakis et al 2016.

TCGA 2012 publications only sequenced 189 non-hypermutated tumors, and Yaeger's study focussed on metastatic tumors, which may have contributed to differences in the mutational frequency.

9. The regression model used to evaluate differences in mutation status between tumor stage, tumor site, and sex lacks clarity, as it is unclear how these variables were corrected for in the model.

Please see the revised Methods section. In this section (and in the tables presenting the results for these analyses), we have attempted to make more clear that these analyses were all adjusted for MSI status only. We also now specify the referent categories used in each analysis.

10. Figure 3 does not convey any novel information.

Figure 3 is now moved to the supplement (Supplementary Figure 4).

Reviewer #2 (Remarks to the Author):

Zaidi et al. perform targeted sequencing for a panel of 205 colorectal cancer-associated genes in >2000 colorectal tumours to identify a number of recurrently mutated genes that had not met significance thresholds and to identify a number of genes that confer a favourable prognosis when mutated. They also show that approximately 1/3 hypermutated tumours that ought to benefit from checkpoint inhibitors do not have detectable micro satellite instability or POLE mutations and so might be missed by existing clinical assays. Previous studies such as MSK-IMPACT had instead termed any hypermutated samples without POLE mutations as 'MSI-H/hypermutated', which was likely misleading.

This paper should be viewed in context of the recent MSK-IMPACT study which performed amplicon-based sequencing on 1134 mostly metastatic colorectal tumours. They were able to determine mutations, copy-number alterations and DNA-level rearrangements for a subset of genes.

In the MSK study, 1,134 tumors were sequenced using the hybrid-capture method to detect copy number alterations and select genomic rearrangements. Our Amplicon based sequencing was not designed to capture genomic rearrangements. We created a compact panel to reduce the cost of sequencing to generate a large dataset. From the FFPE DNA, we were not able to accurately call copy number alterations. These are the limitations of our study which we have discussed in the manuscript.

The main differences between Zaidi et al.'s study and MSK-IMPACT is that a) cohort size is larger thus providing enhanced statistical power to identify recurrent mutations and b) Zaidi et al. only report SNVs and indels, c) Zaidi et al. superimpose survival data (although similar data has been reported by the TCGA project).

We like to point out that no publication using the TCGA data (including the Nature 2012 publication) reported on survival. We had hoped to include TCGA in our survival analysis to improve statistical power; however, we learned that TCGA only has data on overall survival but not CRC-specific survival. As overall survival is a combined measure of many causes using this would attenuate or obscure any association of a gene with CRC-specific survival.

To my mind the authors have not sufficiently followed-up or given due attention to the novel findings of this project and have instead focussed too much on results that have been previously described in substantial detail by TCGA/MSK-IMPACT (e.g. improved survival in HM tumours, patterns of pathway activation).

We generated a large dataset of somatic mutations from over 2,105 well-characterized primary CRC cases to study associations with tumor characteristics and survival. TCGA and the MSK studies have only sequenced 276 and 478 primary tumors, which are substantially smaller. Accordingly, our study is the largest study conducted so far enabling detailed exploration of somatic mutations in CRC, including subclasses of hypermutated tumors, tumor characteristics, and survival. In addition to novel genomic findings, we confirmed previous results and further extended our knowledge of mutated genes and their impact on CRC. We like to point out that despite our very large sample size, we observed limited evidence for genes to be associated with CRC-specific survival and stage. Interestingly when we restricted the analysis to those mutations with putative driver status, we were able to identify an association between *TP53* and CRC-specific survival. As we describe in the manuscript, this finding emphasizes the need for

large sample sizes and detailed functional evaluation of all genetic variants in known and suspected cancer genes, which will be increasingly possible do to the CRISPR scans.

There is a need to expand on:

Biological relevance - recurrent mutations in PRKCI, SPZ1, MUTYH, MAP2K4 (e.g. component of MAPK pathway, mutants cells shown to be sensitive to MEK inhibitors), FETUB

We have now included additional details on the biological relevance of these genes in the Discussion.

Translational relevance - survival association with BMPR2, ZFH3, KMT2B, and RYR1; proportion of hypermutated tumours probably missed by existing assays (e.g. speculate on how these tumours must be identified in a cost-effective way).

Since survival associations of these genes were insignificant after adjusting for multiple comparisons, and upon suggestion by Reviewer 1, we have deemphasized these findings in the text.

Specific comments

There are a number of strengths to the study including the size of the cohort, the robust mutation calling strategy using consensus calls from more than one mutation calling algorithm, successful validation of a random subset of mutation calls using Sanger sequencing and well justified selection of hypermutated samples

The major weakness with the study is the lack of copy number alteration data which is a significant problem - recent work has shown that it is predominantly CNV that drives precursor lesion transformation and subclonal evolution in cancers (Cross et al, 2018). Indeed the title of the paper is thus somewhat misleading as it is only the SNV somatic mutation landscape that is described in this paper.

Cross et al 2018 sequenced multiple regions of fresh-frozen tumors, from 9 adenomas and 14 carcinomas including 4 cases with Lynch syndrome. They concluded CNVs as being the dominant drivers of transformation and subclonal evolution. We were not able to determine accurate copy number changes using our amplicon-based approach and DNA obtained from FFPE tissues. The main advantage of our use of the archival FFPE tissues is the availability of detailed survival data from well-characterized cohorts. This large-scale study could not have been made possible with a recent collection of fresh frozen tissues.

Title has been revised to reflect SNVs and indels.

This is a particular problem in 2 sections:

1. Page 11 the section ascribing the residual transcriptional activity of p53, as only 5% of mutations carried biallelic SNV's. Therefore, a large assumption is made that the single allele mutations were associated with LOH in the other allele. This doesn't appear to have been shown.

Mutations in *TP53* could result in a gain of oncogenic function, accumulation of mutant protein due to reduced degradation, and dominant-negative effect on the wild type protein. As such, heterozygous mutations without the loss of heterozygosity could further reduce the activity of p53 protein from wild type allele by altering the ratios of mutant and wild type p53 proteins and

by generating p53 tetramers with reduced p53 activity. Survival effects of *TP53* mutations with predicted residual activity are in line with these facts. We have now revised Discussion to include these details.

2. Page 15 - the section discussing the favourable prognostic implication of *BMP2* (quoting Voorneveld et al 2014). The latter paper was very clear that the role of BMP signalling is skewed to pro-tumorigenic following loss of *SMAD4*. Although SNV mutation of *SMAD4* is measured, this doesn't account for chrom 18q loss which is the most common way of impacting this pathway in CRC.

Authors agree with the Reviewer. Unfortunately, we were unable to determine loss at 18q. We have revised the Discussion accordingly.

Other comments.

It would be interesting to do further analysis (?CNV, ?methylation) on the 40 or so tumours with very low SNV mutation burden (<5 mutations). What is driving these lesions?

We like to note that we observed 12.5% (n=265) of all tumors had less than 5 mutations. Among these tumors, 78% (n=208) had non-silent mutations in *APC*, *TP53*, *KRAS*, *BRAF*, or *CTNNB1* genes. Further evaluation of tumors for which the driver genes are not known is an interesting idea. This will require substantial resources that are unfortunately not available at this time. Our consortium will continue to seek support to further characterize the collection of observational and case-control studies of colorectal cancer.

The hyper and ultramutated tumours without evidence for *POLE* or *MMR* are interesting and clinically relevant, as discussed in the manuscript. However the reader is left somewhat hanging as to the possible cause of this molecular phenotype. Were other replication fork genes in the targeted panel? Should these tumours undergo more extensive omic analysis to look for novel drivers of mutator phenotype?

In these tumors, we examined mutations in genes involved in DNA replication fork and DNA repair. In a subset of these tumors, mutations are present in *CDK12*, *POLD1*, *NCAPD3*, *RECQL5*, *ATM*, *FAN1*, *ERCC3*, and *XPC*. Please see revised Results and new Supplementary Figure 2.

There is lack of clinicopathological information regarding tumours - age, sample histology e.g. mucinous/signet ring, confirmation that all samples were reviewed by expert histopathological prior to extraction, length of follow-up available for determination of survival data

We now add to the paper "All tumors underwent a pathology review to confirm that the tumor was a primary colorectal carcinoma". However, data on additional histological evaluations are not available to us.

Some of the 'novel' recurrent mutations have been previously described - *TGFBR2* (<https://www.ncbi.nlm.nih.gov/pubmed/17985359>)

Please see revised Discussion with added details about the *TGFBR2* mutations in CRC.

Reviewers' comments:

Reviewer #1 (Remarks to the Author):

In their revised manuscript, Zaidi et al. have now provided the underlying mutational and clinical data and have clarified some aspects of their analysis.

The main concern of this reviewer remains: In their cohort, they find that about 5% of tumors are hypermutated but are neither MSI-H nor affected by driver mutations in POLE or POLD1. If real, this would be an important novel subtype of colorectal cancer that has not been identified or described in other cohorts before (TCGA, Yaeger et al. Giannakis et al., and others) and the description of which would require particular attention in this study. An alternative explanation is that this subtype can, at least in part, be explained by artifacts in the mutation calling pipeline. The authors should first demonstrate that they have ruled out any artifacts and then describe this novel subtype in greater detail.

While the patterns of non-synonymous mutations in genes known to be mutated in colorectal cancer look as expected, the following findings, based on a quick and superficial analysis of the newly provided supplemental data, hint at possible issues with the mutation calling pipeline: 1) Many of these hypermutated / MSS / non-POLE tumors have an unusually high intronic / non-coding mutation burden (which of course could be interesting biology as well). 2) The gene RYR1 contains a suspicious mutational hotspot at amino acid 4469 (E4469A). Is this a real mutation or an artifact? 3) 10 samples have a novel, recurrent frameshift mutation in BRAF, although that could be a previously undetected passenger mutation in MSI-H tumors. But these are just a few small suggestions, and there are many other ways to explore the data.

In summary: In this reviewer's opinion, the authors need to further explore the interesting subtype that is characterized by an increased mutation burden but an absence of the MSI-H phenotype and exonuclease mutations in POLE. This subtype is either biologically novel and interesting or can be explained by issues with the variant data.

Reviewer #2 (Remarks to the Author):

Strengths are as before - Good mutation detection strategy e.g. validation of reads

Some ongoing issues

HM/NHM status as a confounding factor - it is unclear whether the authors have adjusted this analysis to minimise the possibility that hypermutated/non-hypermutated status acts as a confounding factor, as the other reviewer has pointed out this may explain the full extent of the survival association with P53. Presumably this analysis should be performed as a multivariate analysis incorporating mutation status + NHM/HM status + stage + grade + age + sex to look for independent predictors of survival. I believe the authors analysis currently only incorporates age + sex + study.

This issue has been raised by reviewer 1 and I do not feel that the manuscript has been appropriately updated to reflect how tentative these findings of survival benefits are

There is a discrepancy between "in NHM tumours, 77% of tumours have mutations in the Wnt/beta-catenin pathway" (line 140) and "99% of NHM tumours harboured truncating mutations" in APC, please can the authors explain this discrepancy as I believe the first version sent to review referred to a frequency of 62.7% in NHM tumours

Minor issues

Line 32 "comprehensive profiling" - in 2019 a truly 'comprehensive profiling' should include CNAs +/- epigenetic changes

Line 70 - "NHM" has not been defined as non-hypermuted at this point in the manuscript

"As our study includes almost 10 times the sample size of the first TCGA paper (Nature 2012), our study can provide further insights given improved statistical power." This is misleading as the TCGA have published results of their complete cohort (around 600 patients) in Cancer Discovery last year (Grasso et al., 2018).

"We also like to note that we have information on CRC-specific survival, while TCGA has only data on overall survival, which tends to weaken associations between mutated genes and survival" + "As overall survival is a combined measure of many causes using this would attenuate or obscure any association of a gene with CRC-specific survival."
What is the evidence for this (no reference given)?

"We like to point out that no publication using the TCGA data (including the Nature 2012 publication) reported on survival." This is slightly misleading - <https://www.ncbi.nlm.nih.gov/pubmed/29625055>.

Reviewer #1 (Remarks to the Author):

In their revised manuscript, Zaidi et al. have now provided the underlying mutational and clinical data and have clarified some aspects of their analysis.

The main concern of this reviewer remains: In their cohort, they find that about 5% of tumors are hypermutated but are neither MSI-H nor affected by driver mutations in *POLE* or *POLD1*. If real, this would be an important novel subtype of colorectal cancer that has not been identified or described in other cohorts before (TCGA, Yaeger et al. Giannakis et al., and others) and the description of which would require particular attention in this study. An alternative explanation is that this subtype can, at least in part, be explained by artifacts in the mutation calling pipeline. The authors should first demonstrate that they have ruled out any artifacts and then describe this novel subtype in greater detail.

To address this helpful comment we have conducted another round of validation by Sanger sequencing specifically for these 3% of tumors that are hypermutated tumors but neither MSI-H nor affected by non-silent mutations in *POLE* or *POLD1*. We sequenced a total of 63 mutations and confirmed that 95.2% of these calls are confirmed mutants. We now have extended the description of this subset of CRC tumors, which comprises 3% of all CRC tumors. Please see the revised text in the Results, Discussion, and Methods section, where we have provided additional details about these tumors.

While the patterns of non-synonymous mutations in genes known to be mutated in colorectal cancer look as expected, the following findings, based on a quick and superficial analysis of the newly provided supplemental data, hint at possible issues with the mutation calling pipeline: 1) Many of these hypermutated / MSS / non-*POLE* tumors have an unusually high intronic / non-coding mutation burden (which of course could be interesting biology as well). 2) The gene *RYR1* contains a suspicious mutational hotspot at amino acid 4469 (E4469A). Is this a real mutation or an artifact? 3) 10 samples have a novel, recurrent frameshift mutation in *BRAF*, although that could be a previously undetected passenger mutation in MSI-H tumors. But these are just a few small suggestions, and there are many other ways to explore the data.

1) We analyzed the frequency of the types of mutations found in hypermutated tumors. The frequency distribution of the different types, such as intronic, exonic, and UTR mutations are similar among tumors with MSI and MSS tumors with or without *POLE/POLD1* mutations. Please see the new Supplementary Figure 7.

2) This artifact in *RYR1* gene has now been removed from our dataset. See revised Supplementary Table S1.variant calls.

3) The novel recurrent frameshift mutation in *BRAF* was present in 9 tumors with MSI in a region with G repeats. All mutations in these 9 tumors were confirmed by Sanger sequencing.

In summary: In this reviewer's opinion, the authors need to further explore the interesting subtype that is characterized by an increased mutation burden but an absence of the MSI-H phenotype and exonuclease mutations in POLE. This subtype is either biologically novel and interesting or can be explained by issues with the variant data.

We have now included details of this subtype in the manuscript. Please see the revised text in the Results, Discussion, and Methods sections.

Reviewer #2 (Remarks to the Author):

Strengths are as before - Good mutation detection strategy e.g. validation of reads

Some ongoing issues

HM/NHM status as a confounding factor - it is unclear whether the authors have adjusted this analysis to minimise the possibility that hypermutated/non-hypermutated status acts as a confounding factor, as the other reviewer has pointed out this may explain the full extent of the survival association with P53. Presumably this analysis should be performed as a multivariate analysis incorporating mutation status + NHM/HM status + stage + grade + age + sex to look for independent predictors of survival. I believe the authors analysis currently only incorporates age + sex + study.

We appreciate the reviewer's comment that HM/NHM status is an important consideration for our survival analyses. Before including this variable in our analytic model as a confounder, we felt it most important to consider its possible role as an effect modifier given the substantial differences in mutational patterns by HM/NHM status. In some of our analyses, it is quite clear that HM/NHM status is an effect modifier as results for survival differ substantially by HM/NHM status (i.e., *ZFH3*, *RYR1*) - thus making it inappropriate to combine these and adjust for HM/NHM status in our overall model for those particular analyses but instead show results separately by HM/NHM status. In other analyses, it is less clear as to whether or not HM/NHM status is an effect modifier or confounder. We have now conducted additional analyses adjusting for HM/NHM status in those instances, and report on HM/NHM-adjusted analyses when appropriate in Tables 1 and 2.

We also respond to the reviewer's suggestion to adjust for stage at diagnosis in our multivariate analyses. However, given that stage at diagnosis is plausibly a consequence of some of the somatic mutations we are investigating in these analyses, stage cannot be considered as a confounder. That is, part of the effect of these mutations on survival may likely operate via an effect on stage at diagnosis. As such, we now report on how stage adjustment impacted our results in the text of the manuscript as it can help interpret how the mutated gene may impact survival, but do not provide stage-adjusted results in our analytic tables as this could be easily misinterpreted.

This issue has been raised by reviewer 1 and I do not feel that the manuscript has been appropriately updated to reflect how tentative these findings of survival benefits are

We have attempted to downplay our conclusions regarding our survival findings in light of this reviewer's comment.

There is a discrepancy between "in NHM tumours, 77% of tumours have mutations in the Wnt/beta-catenin pathway" (line 140) and "99% of NHM tumours harboured truncating mutations" in APC, please can the authors explain this discrepancy as I believe the first version sent to review referred to a frequency of 62.7% in NHM tumours

We like to note that both statements are correct as one is referring to the subset with APC mutations:

Among APC mutated tumors, 99% of NHM and 85% of HM tumors harbored truncating mutations occurring within the first 1,600 codons, for which truncating mutations are predicted to have the most functional consequences.

In this statement, we are pointing out that in tumors with APC mutations, most occur as truncations within the first 1,600 amino acids. These mutations are of most functional consequences, as mentioned by Vogelstein in his publication (Science 2013).

Minor issues

Line 32 "comprehensive profiling" - in 2019 a truly 'comprehensive profiling' should include CNAs +/- epigenetic changes

We deleted the word comprehensive.

Line 70 - "NHM" has not been defined as non-hypermutated at this point in the manuscript.

We provide a definition for NHM.

"As our study includes almost 10 times the sample size of the first TCGA paper (Nature 2012), our study can provide further insights given improved statistical power." This is misleading as the TCGA have published results of their complete cohort (around 600 patients) in Cancer Discovery last year (Grasso et al., 2018).

The reviewer is correct that our sample size is only about 3 times larger. As this statement was made in response to the reviewer's comment but not included in the manuscript itself we did not make changes to the manuscript

“We also like to note that we have information on CRC-specific survival, while TCGA has only data on overall survival, which tends to weaken associations between mutated genes and survival” + “As overall survival is a combined measure of many causes using this would attenuate or obscure any association of a gene with CRC-specific survival.”

What is the evidence for this (no reference given)?

We regret any confusion this statement may have caused in our previous response to the Reviewer. What we intend to convey here is that CRC-specific survival is a more sensitive, biologically plausible outcome of interest for the somatic alterations under study. Analyses of overall survival combine CRC-specific deaths and deaths that could not plausibly be impacted by the presence of the somatic mutations under study (e.g., heart disease, motor vehicle fatalities). Thus, effect estimates that are specific to associations with CRC-specific deaths are likely to be diluted in analyses of overall survival.

“We like to point out that no publication using the TCGA data (including the Nature 2012 publication) reported on survival.” This is slightly misleading - <https://www.ncbi.nlm.nih.gov/pubmed/29625055>.

We agree with the Reviewer that some survival analysis has been conducted for TCGA. We retract our comment to the Reviewer in our response to the revision.

Reviewers' comments:

Reviewer #1 (Remarks to the Author):

In this revision, Zaidi et al. have looked more closely at the potentially novel subtype of CRC that is hypermutated, MSS, but not explained by any obvious mechanism.

While it is interesting to see that the mutations in the hypermutated MSS cases that do not harbor a POLE exonuclease can be validated by Sanger sequencing, this raises further questions that were unfortunately not addressed in this manuscript:

1) In addition to gene mutations in DNA repair genes (which the authors looked at), is this subtype different in any other way? Most importantly, did the authors perform a mutational signature analysis that could hint at the underlying mechanism?

2) Is this novel subtype exclusive to this particular data set? If the same mutation calling methods were applied to the TCGA data set, would this subtype emerge as well?

3) Or is all of this possibly due to a data issue? Are these mutations introduced into these samples some other way? Are these older FFPE samples? Are the authors using all commonly used variant filters that remove sequencing artifacts (several are listed in methods, but are there others, including 8OxoG)? See this paper for examples:
<https://www.sciencedirect.com/science/article/pii/S2405471218300966>

Reviewer #2 (Remarks to the Author):

I am happy that the authors have now responded satisfactorily to my previous concerns

Reviewer #3 (Remarks to the Author):

I was asked to review this manuscript, which has undergone multiple revisions for Nature Communications, and specifically comment on the ongoing concerns of Reviewer #1.

Overall, this is a nice descriptive manuscript with deep, paired tumor-normal sequencing of a large dataset of CRCs using a 205 gene targeted panel. The authors confirm known mutations of CRC and confirm the association between high TMB with improved prognosis.

Comments for Reviewer #1:

Reviewer #1 suggests that the authors should further examine a subset of tumors that are hypermutated by the authors' definition but lack POLE/POLD1 mutation or MSI-H phenotype. While I agree that examining this subtype would be interesting, this is beyond the scope of the authors' original intent. I suspect that this small group is comprised of a combination of (1) CRCs with isolated MSH6 deficiency, (2) the "long-tail" of MSS tumors with increased TMB, and (3) CRCs with other mechanisms of hypermutation such as MUTYH-associated polyposis. Figuring out the mechanism of each individual cancer would not add to the main conclusions of the current manuscript.

Additional Major Comment:

I agree with comments of Reviewer #2 that hypermutation status confounds many aspects of the analysis. Namely, many of the mutations in HM specimens are expected to be passenger mutations and may not contribute to pathogenesis.

For example, there is a high likelihood that the associations between BMP2, ZFX3, KMT2B, and

RYR1 and survival are false discoveries. Mutations in any these genes may be a surrogate marker of hypermutation, which is not accounted for in the statistical analysis. Note that RYR1 is a large protein with >5000 amino acids and therefore can be more susceptible to somatic mutations in the setting of HM for that reason alone.

This statement is supported by the observation that mutation in each gene is associated with improved (rather than decreased) survival, and the association is no longer significant when HM/NHM is considered. This finding should be excluded from the paper, since it has the potential to confuse the scientific literature and lead others to perform experiments based on a false or misunderstood premise.

Similarly in Table 3, most genes are more frequently mutated in the right colon, which is highly enriched for CRCs with HM. The data is said to have been adjusted for MSI, but it should also be adjusted for HM.

In general, the authors should consider using more sophisticated statistical models for the gene-specific analyses that account for the mutational burden of each individual cancer.

Additional Minor Comments:

Line 136: The authors imply that mutations in MMR genes cause MSI phenotype in 46% of MSI CRCs. This sentence should be removed or clarified with deeper analysis. Abundant literature evidence show that about 80% of CRCs with MSI are caused by MLH1 hypermethylation. Most of mutations in MMR genes in the authors' dataset are expected to be due to incidental passenger mutations from to high mutational burden of MSI-H cancers. I see this frequently in my clinical NGS practice. This can be confirmed, if the authors choose, with MLH1 promoter methylation and MMR gene expression by immunohistochemistry (common tests the patients may have had as part of their routine clinical care).

Line 141: The frequency of specific POLE and POLD1 mutations (for example, P286R) should be stated in the main section of the manuscript. The manuscript implies that all POLE and POLD1 exonuclease mutations identified are pathogenic, which may be unintended or inaccurate.

Figure 1A: The "green" and "aqua" datapoints are nearly indistinguishable. To the left of the dotted line, >80% of the events look orange to me, but closer examination of 1B shows that less than half of these cases actually have MSI. The way that black dotted lines connect 1A and 1B suggest that 1B is an expanded view of 1A; however, 1A and 1B have cases listed in different order.

Perhaps the authors can work with an illustrator at Nature to ensure this figure accurately reflects the data for the reader.

Responses in blue

Reviewers' comments

Reviewer #1 (Remarks to the Author):

In this revision, Zaidi et al. have looked more closely at the potentially novel subtype of CRC that is hypermutated, MSS, but not explained by any obvious mechanism.

While it is interesting to see that the mutations in the hypermutated MSS cases that do not harbor a POLE exonuclease can be validated by Sanger sequencing, this raises further questions that were unfortunately not addressed in this manuscript:

1) In addition to gene mutations in DNA repair genes (which the authors looked at), is this subtype different in any other way? Most importantly, did the authors perform a mutational signature analysis that could hint at the underlying mechanism?

Thank you for the thoughtful comment. As suggested, we have conducted a mutational signature analysis by estimating the fraction of previously described mutational signatures using simulated annealing, as previously described by Huang et al. 2017. *Bioinformatics*;34(2), 330-337. We base this on the latest COSMIC V3 signatures as described by Alexandrov et al. *Nature*. 2020;578(7793):94–101 and described in more detail here: <https://cancer.sanger.ac.uk/cosmic/signatures>. First, we show that we are able to observe the expected mutational signatures for colorectal cancer in our data set: single-base substitution (SBS) signature 1, 6, 10a and 10b (see Figure 1, below). Consistent with previous observations, age at diagnosis is positively associated with SBS1 (p -value = 4.8×10^{-5} unadjusted and $4.9 \times 10^{-$

2 adjusted). Furthermore, as predicted, cases that are hypermutated (HM) and microsatellite instable (MSI) (GP3 in Figure 1) have a larger fraction of SBS6 and cases that are HM, microsatellite stable (MSS) and have *POLE/POLD1* mutations (GP2 in Figure 1) have a higher proportion of SBS10a and SBS10b. This also shows that the subset of cases that are HM-MSS with no *POLE/POLD1* mutations (GP4 in Figure 1) have a higher proportion of SBS6, SBS10a, and SBS10b. Furthermore, we explored other signatures that are associated with DNA repair capacity. Except for SBS30, all mutational signatures associated with DNA repair capacity were higher in the three HM subsets (GP2, GP3, and GP4), including the HM-MSS-non-*POLE/POLD1* mutations (GP4). This suggests that impaired DNA repair mechanisms likely explain the large number of mutations in the case subset of HM-MSS-non-*POLE/POLD1* mutations (GP4). While these data provide more insights, it is important to keep in mind that the targeted sequencing approach of 1.2Mb is suboptimal to define mutational signatures, which can be best done using whole genome sequencing data. Given this and the comment from reviewer 3, we did not include these data in the manuscript. However, if the editor and reviewers suggest that we should add these data we would be happy to do so (probably most would be added in the supplement).

2) Is this novel subtype exclusive to this particular data set? If the same mutation calling methods were applied to the TCGA data set, would this subtype emerge as well?

We explored this question in TCGA and found one case in the TCGA data. We like to point out that due to the sizable amount of tumor tissue needed in TCGA the sample selection for TCGA cases likely differed from the other studies.

3) Or is all of this possibly due to a data issue? Are these mutations introduced into these samples some other way? Are these older FFPE samples? Are the authors using all commonly used variant filters that remove sequencing artifacts (several are listed in methods, but are there others, including 8OxoG)? See this paper for examples:

<https://www.sciencedirect.com/science/article/pii/S2405471218300966>

To address this helpful question, we compare the overall mutational type frequency of all mutations and the 3' and 5' prime base in our sample with the overall mutational type frequency from TCGA colorectal cancer samples (see Figure 2 below). To ensure a valid comparison, we restricted the TCGA data to the targeted gene content of our panel. Importantly as can be seen in Figure 2, the overall mutational type frequency is very similar for TCGA (based on fresh frozen) and our study, suggesting no evidence for FFPE artifacts in our data.

Furthermore, we now provide the quality control estimates, including the age of the samples and OxoG Q score separately for NHM (group 1), HM-MSS-*POLE/POLD1* mutated (group 2), HM-MSI (group 3), and HM-MSS-non-*POLE/POLD1* (group 4) and a description of the assessment in the Supplementary Text document (Section 4). As shown in the Table in Section 4 of the Supplemental Text, the four groups show very similar quality control values, suggesting no major differences that could explain the observed mutational profile of HM-MSS-non-*POLE/POLD1* (group 4). This indicates that FFPE samples can be utilized for targeted deep sequencing as also commonly done in the clinic given the wide availability of FFPE tissue

compared with fresh frozen tissue. Similarly, the vast majority of well-characterized observational studies, with long-term follow up data need to rely on FFPE tissue as fresh frozen is not available.

Reviewer #2 (Remarks to the Author):

I am happy that the authors have now responded satisfactorily to my previous concerns

Reviewer #3 (Remarks to the Author):

I was asked to review this manuscript, which has undergone multiple revisions for Nature Communications, and specifically comment on the ongoing concerns of Reviewer #1.

Overall, this is a nice descriptive manuscript with deep, paired tumor-normal sequencing of a large dataset of CRCs using a 205 gene targeted panel. The authors confirm known mutations of CRC and confirm the association between high TMB with improved prognosis.

We thank the reviewer for the positive response. In addition to these findings, we identified several novel driver genes, we also observed that TP53 is associated with CRC-specific survival, particularly when accounting for the residual activity. Besides from this finding for TP53, we found limited evidence for an association of specific genes or pathways with CRC-specific survival (or overall survival), despite our sizable sample size and long-term follow up. This finding itself is important and helps to interpret many previous publications that focussed on small numbers of genes in smaller numbers of cases often reported significant associations between somatically mutated genes and survival.

Comments for Reviewer #1:

Reviewer #1 suggests that the authors should further examine a subset of tumors that are hypermutated by the authors' definition but lack POLE/POLD1 mutation or MSI-H phenotype. While I agree that examining this subtype would be interesting, this is beyond the scope of the authors' original intent. I suspect that this small group is comprised of a combination of (1) CRCs with isolated MSH6 deficiency, (2) the "long-tail" of MSS tumors with increased TMB, and (3) CRCs with other mechanisms of hypermutation such as MUTYH-associated polyposis. Figuring out the mechanism of each individual cancer would not add to the main conclusions of the current manuscript.

In light of this helpful comment, we have not further expanded on the description of this small fraction of cases within the manuscript; however, we address reviewer 1's comments in this response.

Additional Major Comment:

I agree with comments of Reviewer #2 that hypermutation status confounds many aspects of the analysis. Namely, many of the mutations in HM specimens are expected to be passenger mutations and may not contribute to pathogenesis.

For example, there is a high likelihood that the associations between *BMP2*, *ZFX3*, *KMT2B*, and *RYR1* and survival are false discoveries. Mutations in any these genes may be a surrogate marker of hypermutation, which is not accounted for in the statistical analysis. Note that *RYR1* is a large protein with >5000 amino acids and therefore can be more susceptible to somatic mutations in the setting of HM for that reason alone.

This statement is supported by the observation that mutation in each gene is associated with improved (rather than decreased) survival, and the association is no longer significant when HM/NHM is considered. This finding should be excluded from the paper, since it has the potential to confuse the scientific literature and lead others to perform experiments based on a false or misunderstood premise.

We appreciate this thoughtful comment and understand that the observed effect modification by HM can easily be misinterpreted by the reader. We also acknowledge that the interactions are not surpassing stringent Bonferroni adjustment (although the association with survival do) and, therefore, in line with the reviewer's comment, we no longer highlight the survival analysis for *BMP2*, *ZFX3*, *KMT2B*, and *RYR1*. Table 1 has been removed, and other tables are renumbered.

Similarly in Table 3, most genes are more frequently mutated in the right colon, which is highly enriched for CRCs with HM. The data is said to have been adjusted for MSI, but it should also be adjusted for HM.

We have revised the analysis and now adjust also for the mutational burden as measured by the number of mutations. Please see the revised Table 2 (which was Table 3 in the previous submission).

In general, the authors should consider using more sophisticated statistical models for the gene-specific analyses that account for the mutational burden of each individual cancer.

To address this helpful comment, we have now adjusted all analyses also for the mutational burden as measured by the number of mutations.

Additional Minor Comments:

Line 136: The authors imply that mutations in MMR genes cause MSI phenotype in 46% of MSI CRCs. This sentence should be removed or clarified with deeper analysis. Abundant literature evidence show that about 80% of CRCs with MSI are caused by MLH1 hypermethylation. Most of mutations in MMR genes in the authors' dataset are expected to be due to incidental

passenger mutations from to high mutational burden of MSI-H cancers. I see this frequently in my clinical NGS practice. This can be confirmed, if the authors choose, with MLH1 promoter methylation and MMR gene expression by immunohistochemistry (common tests the patients may have had as part of their routine clinical care).

The authors agree with the Reviewer. We have removed the sentence from the main text.

Line 141: The frequency of specific POLE and POLD1 mutations (for example, P286R) should be stated in the main section of the manuscript. The manuscript implies that all POLE and POLD1 exonuclease mutations identified are pathogenic, which may be unintended or inaccurate.

We have now described the frequencies of recurrent mutations in the exonuclease domains of *POLE* and *POLD1* in the Results section.

Figure 1A: The "green" and "aqua" datapoints are nearly indistinguishable. To the left of the dotted line, >80% of the events look orange to me, but closer examination of 1B shows that less than half of these cases actually have MSI. The way that black dotted lines connect 1A and 1B suggest that 1B is an expanded view of 1A; however, 1A and 1B have cases listed in different order.

Perhaps the authors can work with an illustrator at Nature to ensure this figure accurately reflects the data for the reader.

To better visualize data, we have changed the colors of data points, decreased the size of the points, and jittered them for easier visualization of the overlapping data. We have removed the dotted lines to make Panels A and B as stand-alone figures. To closely resemble, listing in Panel A, we have reordered the tumors in Panel B. Please note the tumors in Panel A are sorted from the highest to the lowest number of mutations. Whereas, in Panel B, tumors are first grouped on their MSI, *POLE* mutation, and MMR statuses and then sorted from highest to the lowest number of mutations.

SBS1 is correlated with age

SBS6 is associated with defective DNA mismatch repair and is found in microsatellite unstable tumors

SBS10a is found with polymerase epsilon exonuclease domain mutations

SBS10b is found with polymerase epsilon exonuclease domain mutations

SBS14 is concurrent with polymerase epsilon mutation and defective DNA mismatch repair

SBS15 is associated with defective DNA mismatch repair

SBS20 is concurrent with *POLD1* mutations and defective DNA mismatch repair

SBS21 occurs with DNA mismatch repair deficiency

SBS26 occurs with Defective DNA mismatch repair

SBS30 occurs with defective base excision repair, including DNA damage due to reactive oxygen species, due to biallelic germline or somatic *NTHL1* mutations

Figure 1. Mutational Signatures for single-base substitution (SBS) stratified by hypermutation, MSI, and *POLE* and *POLD1* mutational status. The red % at the top of the violin plot shows the % of cases in the subgroup of cases with a non-zero signature. The first four signatures (SBS1, SBS6, SBS10a, and SBS10b) are commonly observed in colorectal cancer. The remaining signatures are linked to DNA repair capacity limited to those that occurred frequently enough to derive violin plots (SBS14, SBS15, SBS20, SBS21, SBS26, SBS30).

Figure 2. Comparing mutation type frequency distributions between TCGA and our targeted sequencing data (V3) after applying our filters to both data sets. TCGA colorectal tumor data was restricted to our targeted panel subsetting from whole exome sequencing data. Mutation types are categorized into six mutation substitution subtypes: C>A, C>G, C>T, T>A, T>C, and T>G, and their immediately 5' and 3' bases, e.g. AxA.

REVIEWERS' COMMENTS:

Reviewer #3 (Remarks to the Author):

The authors have addressed my concerns. This article is a useful analysis of mutations identified in a large cohort of colorectal cancers.

Fei Dong, MD